# Constructing sulfur and oxygen super-coordinated main-group electrocatalysts for selective and cumulative $H_2O_2$ production

Xiao Zhou [1,3], Yuan Min[1,3], Changming Zhao[2,3], Cai Chen[2], Ming-Kun Ke[1], Shi-Lin Xu[1], Jie-Jie Chen [1], Yuen Wu [2] ✉ & Han-Qing Yu [1] ✉

Direct electrosynthesis of hydrogen peroxide ($H_2O_2$) via the two-electron oxygen reduction reaction presents a burgeoning alternative to the conventional energy-intensive anthraquinone process for on-site applications. Nevertheless, its adoption is currently hindered by inferior $H_2O_2$ selectivity and diminished $H_2O_2$ yield induced by consecutive $H_2O_2$ reduction or Fenton reactions. Herein, guided by theoretical calculations, we endeavor to overcome this challenge by activating a main-group Pb single-atom catalyst via a local micro-environment engineering strategy employing a sulfur and oxygen super-coordinated structure. The main-group catalyst, synthesized using a carbon dot-assisted pyrolysis technique, displays an industrial current density reaching 400 mA cm$^{-2}$ and elevated accumulated $H_2O_2$ concentrations (1358 mM) with remarkable Faradaic efficiencies. Both experimental results and theoretical simulations elucidate that S and O super-coordination directs a fraction of electrons from the main-group Pb sites to the coordinated oxygen atoms, consequently optimizing the *OOH binding energy and augmenting the 2e$^-$ oxygen reduction activity. This work unveils novel avenues for mitigating the production-depletion challenge in $H_2O_2$ electrosynthesis through the rational design of main-group catalysts.

Hydrogen peroxide ($H_2O_2$) ranks among the most essential and fundamental chemicals, as it is extensively applied in chemical and medical industries, as well as environmental remediation[1–3]. Owing to the persistent growth in directly correlated industrial needs, it is projected that the global $H_2O_2$ market demand will approach approximately 6 million tons by 2024[4]. Presently, over 95% of all $H_2O_2$ is industrially synthesized via the anthraquinone cycling process, which is encumbered by intensive energy consumption, significant organic waste, and safety concerns stemming from the instability of $H_2O_2$ during transport and storage[5,6]. In apparent contrast, the electrochemical two-electron (2e$^-$) oxygen reduction reaction (ORR) offers a more direct pathway for $H_2O_2$ production on-site under ambient conditions[7,8]. Typically, numerous cathode

materials favor the competing 4e$^-$ ORR to $H_2O$ rather than the 2e$^-$ ORR, resulting in diminished $H_2O_2$ yields[1,9]. Noble metal alloy catalysts with isolated reactive sites (e.g., Au-Pd, Pt-Hg, Pd-Hg)[10–12] are discovered to be advantageous for "end-on" adsorption (Pauling-type) of $O_2$, which minimizes O−O bond breaking, facilitating the 2e$^-$ ORR process for $H_2O_2$ production[13,14]. However, the exorbitant cost and resource scarcity of noble metals severely impede their large-scale implementation. Therefore, it is the key to develop high-selectivity and non-noble metal catalysts for $H_2O_2$ electrosynthesis.

Heterogeneous single-atom catalysts (SACs), characterized by isolated active sites and distinctive electronic structures, have been shown to hold considerable promise as materials for energy conversion and chemical transformation[15,16]. In particular, transition-metal

[1]CAS Key Laboratory of Urban Pollutant Conversion, Department of Environmental Science and Engineering, University of Science and Technology of China, Hefei 230026, China. [2]School of Chemistry and Materials Science, University of Science and Technology of China, Hefei 230026, China. [3]These authors contributed equally: Xiao Zhou, Yuan Min, Changming Zhao. ✉e-mail: yuenwu@ustc.edu.cn; hqyu@ustc.edu.cn

SACs with well-defined M-N$_x$ moieties have been recognized as exceptional electrocatalysts for H$_2$O$_2$ production[17–19]. Taking the Co-N$_4$ moiety as an example, wherein tuning the coordination structure and surrounding atomic configuration effectively optimizes the oxygen intermediates and governs the selectivity towards H$_2$O$_2$ synthesis[20,21]. This can be primarily attributed to their unique electronic properties, which exhibit partially occupied $d$ electrons that confer a robust capacity for adsorbing and activating O$_2$, as predicted and elucidated by the $d$-band center theory[22,23]. Regrettably, reactive transition-metal sites also exhibit strong interactions with synthesized H$_2$O$_2$, instigating undesired H$_2$O$_2$ reduction reactions (H$_2$O$_2$RR) within the ORR potential range. Consequently, the generated H$_2$O$_2$ undergoes decomposition, significantly undermining H$_2$O$_2$ yield[24,25]. In addition, transition metal-induced Fenton reactions may contribute to a reduction in H$_2$O$_2$ production and compromise electrode durability[26,27]. To this end, the potent adsorption strength of transition-metal sites to oxygen-containing species may exhaust accumulated H$_2$O$_2$, thereby curtailing overall production. It is reasonable to predict that optimizing 2e$^-$ ORR selectivity for intrinsically less reactive catalysts for H$_2$O$_2$ decomposition may yield higher H$_2$O$_2$ quantities, although this remains a promising yet formidable challenge for H$_2$O$_2$ electrosynthesis. Distinct from $d$-block transition metals, main-group metals with fully occupied $d$-orbitals ($d^{10}$ electronic configuration) are generally deemed catalytically inferior for the electron transfer process of catalytic reaction[28]. Main-group metals should inherently surpass transition metals in terms of inert activity for H$_2$O$_2$RR and Fenton-like reactions, owning to the lack of a combination of empty and occupied host orbitals. Intriguingly, main-group catalysts have been demonstrated as promising candidates for electrocatalytic reactions, boasting enhanced durability by mitigating transition metal-induced Fenton reactions[29–32]. Thus, it should be feasible to concurrently attain selective 2e$^-$ ORR for H$_2$O$_2$ production and alleviate consecutive decomposition through the development of main-group 2e$^-$ ORR SACs.

Motivated by the above analysis, we first predicted the 2e$^-$ ORR process on lead (Pb) SACs with diverse local coordination environments using density functional theory (DFT) calculations. Accordingly, a versatile strategy was devised to regulate the microenvironments of main-group metal sites by meticulously modulating the first coordination with sulfur (S) and oxygen (O). We then fabricated the main-group Pb SACs (Pb SA/OSC) employing a carbon dot-assisted pyrolysis approach, wherein the S/O ratio of the carbon dots was rationally controlled. Advanced microscopy analysis and synchrotron X-ray absorption fine structure spectroscopy corroborated the successful regulation of the coordination configuration of the main-group SACs via S and O dual coordination. The 2e$^-$ ORR performance of the as-fabricated Pb SA/OSC was subsequently assessed utilizing the rotating ring-disk electrode technique. Moreover, to explore the possibility for practical application, Pb SA/OSC was assembled into a functional flow-cell device for H$_2$O$_2$ production at industrial currents. This distinctive main-group Pb catalyst potentially heralds a novel avenue for the exploitation of main-group elements in designing electrochemical catalysts for H$_2$O$_2$ synthesis.

## Results

### Coordination engineering of Pb SA/OSC

We first examined the adsorption of ORR reaction intermediates on Pb single-atom models with varying local coordination environments to gain insights into the 2e$^-$ ORR selectivity employing DFT calculations. Pb-based catalysts were chosen as main-group model catalysts, as Pb materials have been demonstrated to exhibit high activity for diverse electrocatalytic reactions[33–35]. Prior research has indicated that the catalytic reactivities of metal sites are influenced by their coordination environments due to electronic structure alterations[36]. Additionally, it has been reported in the literature that the most common coordination numbers for main-group Pb sites are 4 and 6[37,38]. In this instance,

Pb SA/OSC structure models were constructed by supporting Pb atoms on graphene with various combinations of coordinated S and O atoms (Fig. 1a and Supplementary Fig. 1). The optimized geometry of PbS$_4$O$_2$ exhibits a planar coordination structure with distances of Pb−O (2.5 - 2.6 Å) and Pb−S (2.4 Å) (Supplementary Table 1). In contrast, the distances of Pb−S and Pb−O in the other Pb SACs structures display longer interatomic distances. Accordingly, the adsorption energies of *OOH, *O and *OH on the selected PbS$_4$O$_2$ and other Pb SACs structures were calculated. Correlations between $\Delta G$(*OOH) and the limiting potential of the 2e$^-$ ORR were established to depict scaling relations (Fig. 1b), suggesting that PbS$_4$O$_2$ lies nearest to the apex of the limiting potential volcano, akin to the PtHg$_4$ catalyst[10]. Thus, PbS$_4$O$_2$ affords high selectivity to for H$_2$O$_2$ formation with a low overpotential of 0.006 V. Other Pb sites are situated on either the left or right side of the volcano plots (Supplementary Fig. 2), indicating that the adsorption of *OOH is excessively strong or weak, respectively. To provide a more precise description of the ORR reaction, correlations between the $\Delta G$(*OH) and the limiting potential of the 4e$^-$ ORR were examined. The limiting potential of PbS$_4$O$_2$ (0.53 V) was far away from the equilibrium potential of the 4e$^-$ ORR (1.23 V), insinuating that PbS$_4$O$_2$ is inactive for the reduction of O$_2$ to H$_2$O. Furthermore, the free energy diagram of the ORR reveals that the adsorption of *OOH on PbS$_4$O$_2$ (−0.71 eV) is suitable for the formation and desorption of H$_2$O$_2$, while that on PbS$_4$ (−1.65 eV) is stronger, once again indicating the high selectivity of the 2e$^-$ ORR on the PbS$_4$O$_2$ structure (Fig. 1c).

The adsorption strength of *OOH is recognized as a pivotal descriptors in governing the ORR selectivity of catalysts. Specifically, the binding of *OOH relies upon the electronic energy level of the active site and the corresponding charge transfer at the catalytic interface. Bader charge analysis disclosed that *OOH bears a negative charge with fewer electrons transferred from PbS$_4$O$_2$ (0.58 e$^-$, Fig. 1d) compared to PbS$_4$ (0.65 e$^-$, Fig. 1f), consonant with the calculated adsorption strength of *OOH. Color mapping of the Bader charge on each atom shows that the electrons provided by the Pb site are transferred not only to the adsorbed *OOH, but also to the coordinated O atoms of PbS$_4$O$_2$ (Supplementary Fig. 3). In contrast, electrons of Pb site in PbS$_4$ are not transferred to the coordinated S atoms, resulting in the strongly adsorbed *OOH gaining more electrons. Projected density of states (PDOS) plots exhibits an overlap of the energy levels of the Pb 6$p$ orbitals and the O 2$p$ orbitals of *OOH (Fig. 1e), implying that the adsorption of *OOH is contributed by the Pb 6$p$ orbitals of PbS$_4$. For PbS$_4$O$_2$, no overlapped region was observed near the Fermi level (Fig. 1g), indicating that the Pb 6$p$ orbitals were controlled by the coordinated O atom. The impact of the coordination environments on the conversion of *OOH to H$_2$O$_2$ can be comprehended not only by the weakening of the Pb−OOH* bond, but also by the tightening of the O−O bond in OOH* adsorbed on the Pb site of PbS$_4$O$_2$ (Supplementary Table 2). Therefore, the DFT calculations unveiled that the formation of S and O super-coordinated Pb moieties is the origin of the exceptional 2e$^-$ ORR selectivity of Pb SA/OSC, as a suitable interaction between Pb 6$p$ orbitals and O 2$p$ orbitals gives rise to an optimal adsorption strength of *OOH intermediates.

### Synthesis and characterizations of Pb SA/OSC

To corroborate the theoretical prediction of exceptional 2e$^-$ ORR selectivity of Pb SA/OSC, the main-group single atoms were in-situ anchored onto carbon supports employing a carbon dot-assisted pyrolysis approach. The schematic preparation process of Pb SA/OSC is illustrated in Fig. 2a. S, O-functionalized carbon dots (SCDs) were first prepared by utilizing varying ratios of S/O-containing precursors, furnishing a versatile platform for modulating of the local atomic environments of Pb SACs. Notably, the strong chemical interaction between metal and doped sulfur atoms in SCDs endowed the formation of strong and thermally stable metal-sulfur bonding, markedly

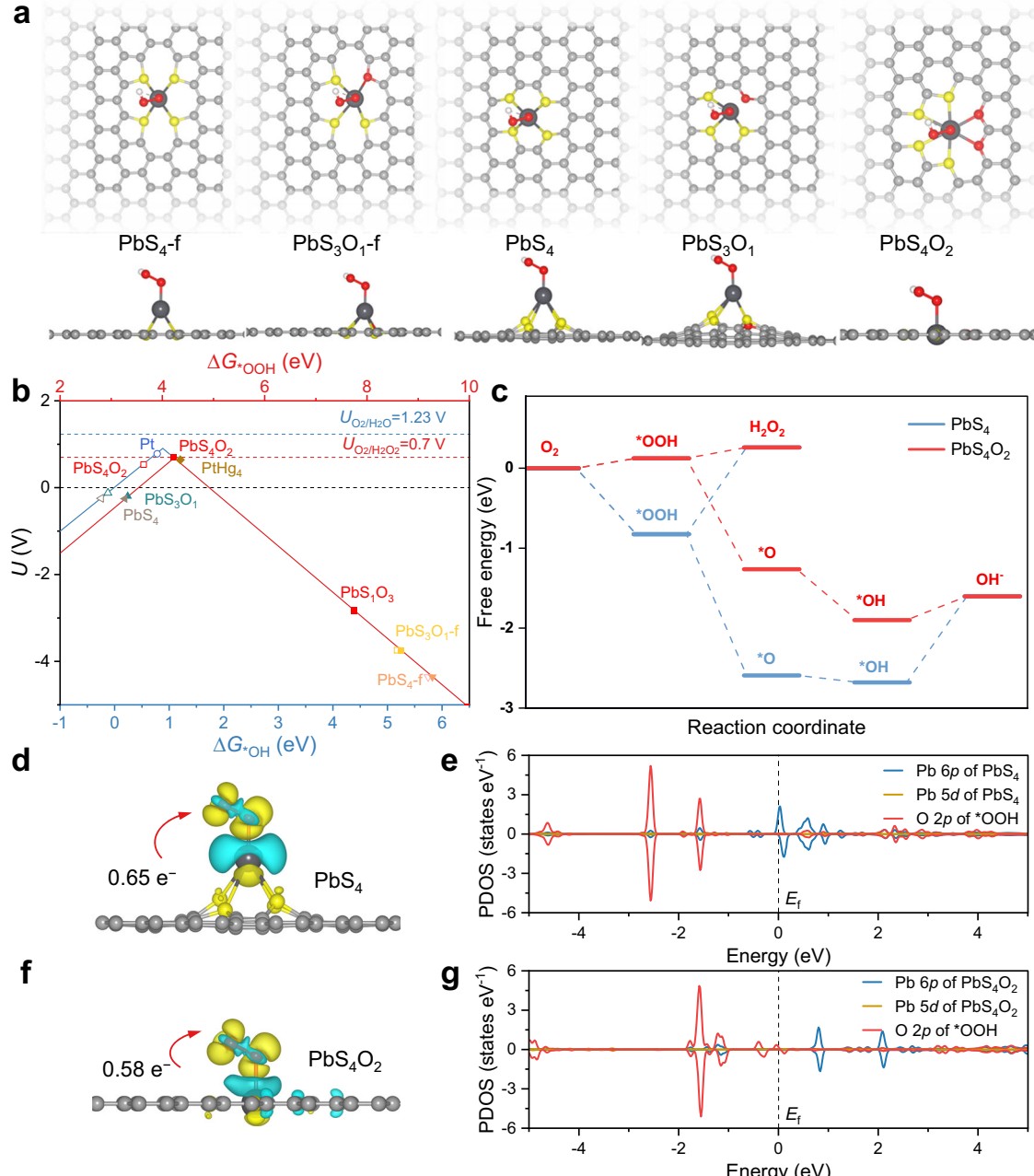

**Fig. 1 | Theoretical ORR activity of Pb SA/OSC. a** Top and side views of *OOH species adsorbed on Pb SA/OSC. Structural motifs considered in the catalyst model of Pb SA/OSC, including in-plane structures of $PbS_4$-f, $PbS_3O_1$-f and $PbS_4O_2$, tetragonal structures of $PbS_4$ and $PbS_3O_1$. Configurations are denoted by the S/O atomic ratio and tpye, with "f" indicating the presence of thiophene-type sulfur. **b** Volcano plots for the $2e^-$ and $4e^-$ ORR on various Pb SA/OSC, Pt and $PtHg_4$[10], with the limiting potential plotted as a function of $\Delta G_{*OH}$ (blue horizontal axis) and $\Delta G_{*OOH}$ (red horizontal axis). The equilibrium potentials of $U_{O2/H2O2}$ and $U_{O2/H2O}$ are 1.23 V and 0.7 V, respectively. **c** Free energy diagram of ORR on $PbS_4$ and $PbS_4O_2$ at an applied potential of zero in alkaline media. **d, f** Differential charge density plots for *OOH species adsorbed on $PbS_4$ and $PbS_4O_2$, with blue and yellow areas representing a loss and gain of electrons, respectively. The isosurfaces were taken at a charge density of $\pm0.002$ e$^-$/bohr³. (e.g.) PDOS plots of the $PbS_4$ and $PbS_4O_2$ catalysts with *OOH.

mitigating the aggregation of metal sites[39,40]. The main-group Pb cations were adequately chelated with the S and O-function groups of SCDs, forming a metal coordination composite, which was assembled on carbon black substrates via π-π interactions[41]. Therefore, the interfacial metal-sulfur bonding between the Pb cations and SCDs effectively anchored and stabilized the main-group sites, proffering an atomically dispersed metal precursor. Then, the main-group Pb SACs with a distinctive S and O dual coordination were obtained via pyrolysis treatment under a protective flowing Ar atmosphere. For comparison, a series of control samples with divergent coordination environments were also synthesized under otherwise identical conditions but employing distinct carbon dot precursors. These were denoted as Pb SA/OC, Pb SA/OSC-2.5 and Pb SA/OSC-7.5, respectively.

Transmission electron microscopy (TEM) images revealed that Pb SA/OSC largely retained the spherical configuration of carbon black, albeit with a marginally larger size (Supplementary Figs. 8a, b). As shown in Fig. 2b, the obtained sample possessed a layered graphene structure epitaxially grown on carbon black substrates, with no discernible Pb-containing nanoparticles detected. Likewise, observations in the high-angle dark-field scanning TEM (HAADF-STEM) image

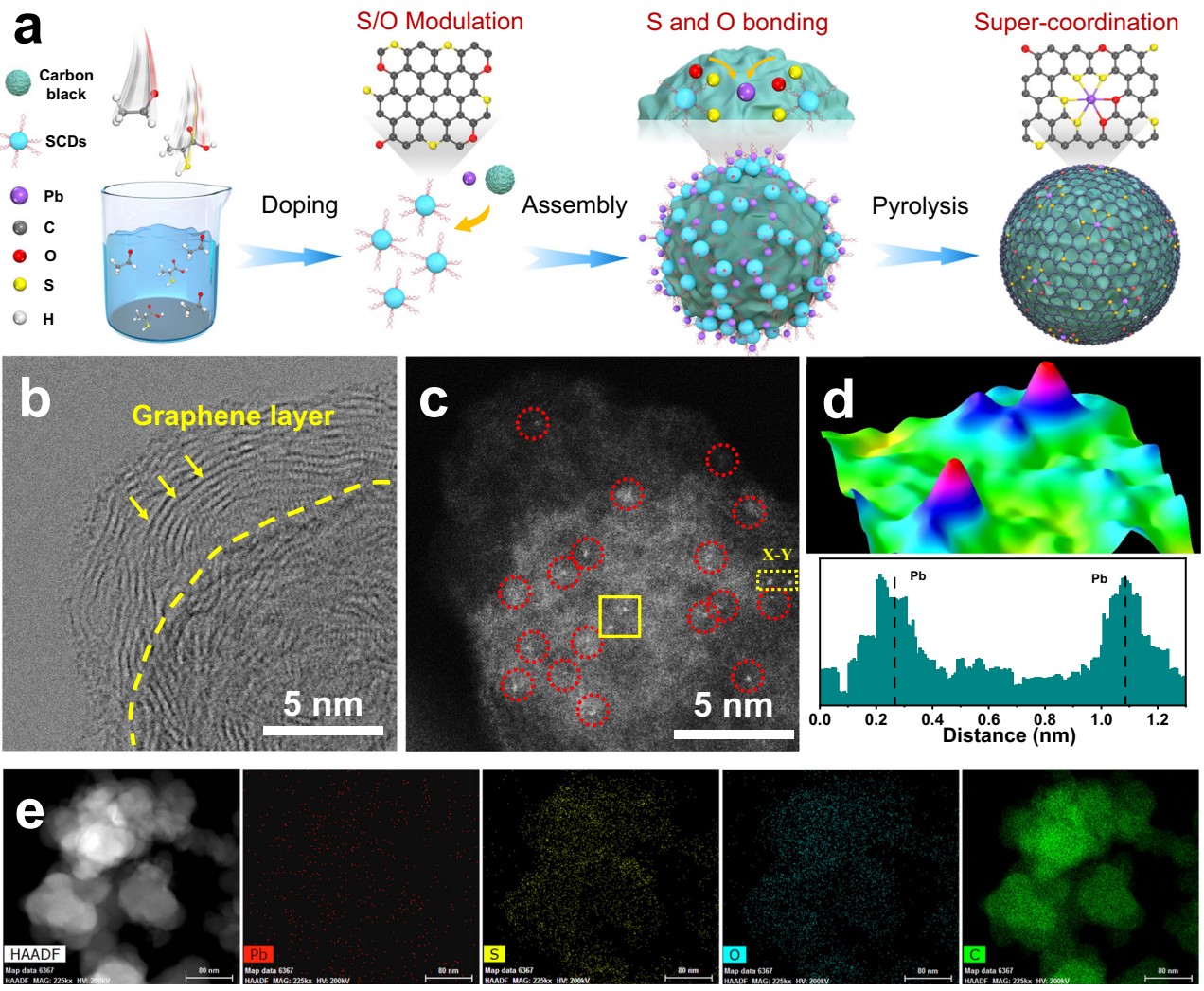

**Fig. 2 | Morphology and structure of Pb SA/OSC. a** Schematic illustration of the preparation of Pb SA/OSC. **b** STEM image and (**c**) AC HAADF-STEM image of Pb SA/OSC. **d** Atom-overlapping Gaussian-function fitting mapping of the square from (**c**), intensity profile along X-Y in (**c**). **e** EDS elemental mapping of Pb SA/OSC, suggesting uniform distributions of Pb, S, O and C elements.

(Supplementary Fig. 8c) corroborated the aforementioned conclusion. Additionally, no characteristic crystal peaks of metallic Pb or PbS was observed in the XRD pattern (Supplementary Fig. 9) of Pb SA/OSC, harmonizing with the prior TEM results. It merits attention that Raman results (Supplementary Fig. 10) disclosed the characteristic D and G bands of conductive carbon materials ($I_D/I_G = 0.9$), implying the presence of copious defects in the SCDs-derived carbon materials, favorable for anchoring isolated metal atoms[42]. The porous structure of Pb SA/OSC was studied by the Brunauer–Emmett–Teller (BET) method, displaying a large BET specific surface area of 739 m$^2$/g (Supplementary Fig. 11). Furthermore, the OSC, Pb SA/OC, Pb SA/OSC-2.5, and Pb SA/OSC-7.5 samples exhibited analogous morphologies, carbon crystallinities and defects, indicating the successful preparation of the main-group Pb SACs (Supplementary Figs. 10, 13–15). The monodispersion of Pb species could be directly monitored by aberration-corrected HAADF-STEM (AC HAADF-STEM). As shown in Fig. 2c and Supplementary Fig. 16, the individual bright dots (labeled with red circles) corresponding to isolated Pb atoms can be unambiguously distinguished from the carbon matrix. The atomic dispersion of Pb in Pb SA/OSC was further substantiated by atom-overlapping Gaussian-function fitting mapping and analysis of intensity profile (Fig. 2d). The energy-dispersive X-ray spectroscopy (EDS) mapping from Fig. 2e indicated that Pb, S, O and C on the supports

were uniformly distributed. Furthermore, the actual content of Pb was quantified as 1.2 wt.% via inductively coupled plasma optical emission spectroscopy (ICP–MS) analysis.

**Analysis of atomic and electronic structure**

To scrutinize the local electronic and atomic structure of the main-group Pb SACs, synchrotron soft X-ray absorption near-edge structure (XANES) was conducted. The S L-edge XANES spectrum (Fig. 3a) of Pb SA/OSC was dominated by three peaks within the range of 163 - 168 eV corresponding to C−S−C coordination species, which concurred with the findings of the SCDs sample[43,44]. This observation compellingly suggested the anchoring of S in the carbon framework, validated by the X-ray photoelectron spectroscopy (XPS) survey results (Supplementary Fig. 17). Meanwhile, the carbon K-edge spectrum (Supplementary Fig. 18) showed four characteristic peaks situated at 286.2 eV (peak A), 288.1 eV (peak $B_1$), 289.3 eV (peak $B_2$) and 293.5 eV (peak C), attributable to the transition of the 1s core electron of carbon into the $\pi^*$ (C = C), $\pi^*$ (C−O/S−C), and $\sigma^*$ (C−C) antibonding states, respectively[45]. Sulfur was further probed by the high-resolution S 2$p$ spectrum. As shown in Fig. 3b, the peak at 168.3 eV was ascribed to the sulfate species (C−SO$_x$), while the two peaks at 165.2 eV and 164.2 eV were associated with C = S−C and C−S−C bonds, respectively[46]. It should be noted that a peak at 163.8 eV corresponding to Pb−S coordination was

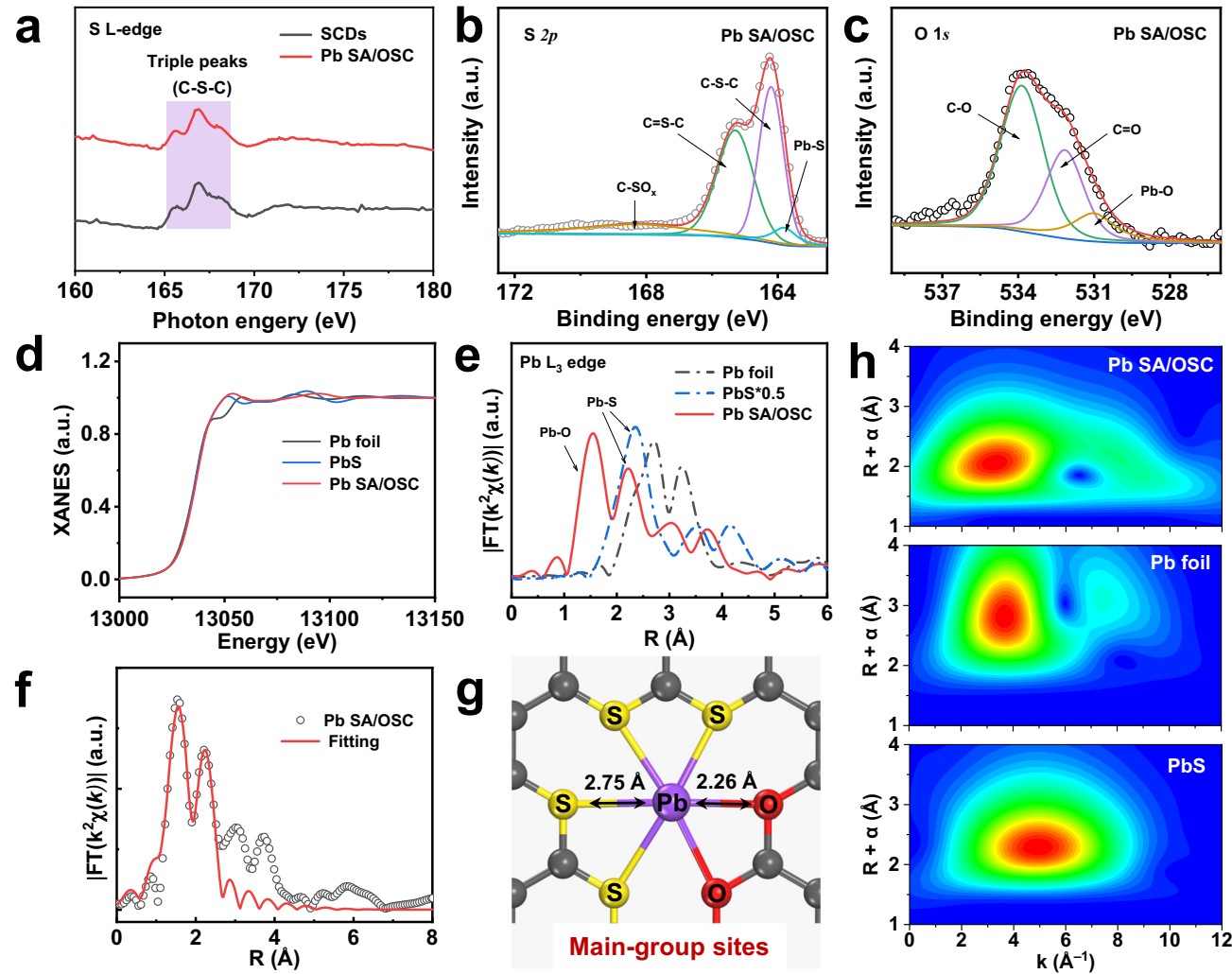

**Fig. 3 | Atomic and electronic structure of the Pb SA/OSC catalyst. a** S L-edge XANES spectra of SCDs and of Pb SA/OSC samples. **b**, **c** S 2p and O 1s XPS spectra of Pb SA/OSC. **d** The normalized Pb L₃-edge XANES spectra of the Pb SA/OSC catalyst and the references (Pb foil and PbS). **e** Pb L₃-edge FT-EXAFS spectra of Pb SA/OSC and reference samples. **f** Corresponding EXAFS fitting curves of Pb SA/OSC in R space. **g** Schematic atomic interface model of Pb SA/OSC. **h** WT-EXAFS plots of Pb SA/OSC, Pb foil and PbS, respectively.

observed. The high-resolution O 1s spectrum can be deconvoluted into C-O, C = O, and Pb-O, indicating the formation of Pb-O bonds (Fig. 3c). These findings insinuated that the atomically dispersed Pb exhibited typical Pb-S and Pb-O dual coordination environments.

Extended X-ray absorption fine structure (EXAFS) was examined to glean detailed structural information for the main-group SACs. As depicted in Fig. 3d, the white line intensity at the Pb L₃ edge of Pb SA/OSC resembled that of PbS, signifying that the Pb atom was featured with an ionic state. The Fourier transform (FT) EXAFS spectra of Pb SA/OSC and the references are illustrated in Fig. 3e. The sample displayed a dominant peak located at about 1.55 Å, predominantly attributed to the scattering of Pb−O coordination. Notably, a peak at about 2.2 Å of Pb SA/OSC was ascribed to Pb-S path scattering, as discerned by the FT-EXAFS spectra compared with PbS. Moreover, the absence of the corresponding feature of the Pb−Pb bond substantiated the atomic dispersion of Pb on the carbon framework. Quantitatively, the coordination configuration of the Pb atom in the Pb SA/OSC catalyst could be acquired via EXAFS fitting, as shown in Fig. 3f. The best-fit result of the EXAFS data in Supplementary Table 6 indicates that the isolated Pb atom was atomically dispersed on the S, O co-doped carbon matrix and likely coordinated within a mixture of Pb-S and Pb-O moieties (denoted as Pb₁-S₄O₂, Fig. 3g). It is worth noting that the averaged

interatomic distances of EXAFS fitting results concurred with the distances of Pb-O and Pb-S in the PbS₄O₂ structure but diverged from the distances of Pb-S and Pb-O in the other Pb SACs structures, substantiating the super-coordinated structure of Pb SA/OSC. Wavelet transform (WT) of Pb L₃-edge EXAFS oscillations was conducted to further corroborate the atomic monodispersity of the Pb species in Pb SA/OSC. From the WT contour plots of Pb foil and PbS standards (Fig. 3h), the intensity maxima at 3.7 and 4.9 Å⁻¹ were most likely associated with Pb−Pb and Pb−S, respectively. In contrast, the WT contour plot of Pb SA/OSC displayed an intensity maximum at 3.2 Å⁻¹, attributable to Pb−S bonding. Hence, the above results compellingly suggested the isolated and super-coordinated feature of Pb species in Pb SA/OSC, confirming that the local atomic environments of Pb single sites were modulated by coordination engineering.

### ORR performance measurement

Differing from the various SACs reported in the literature, the as-fabricated Pb SA/OSC material exhibited a characteristic main-group center and a S, O super-coordinated configuration. Inspired by the distinctive local environment of Pb SA/OSC, the performance of 2e⁻ ORR was subsequently assessed using a standard three-electrode system with rotating ring-disk electrode (RRDE) in O₂-saturated 0.10 M

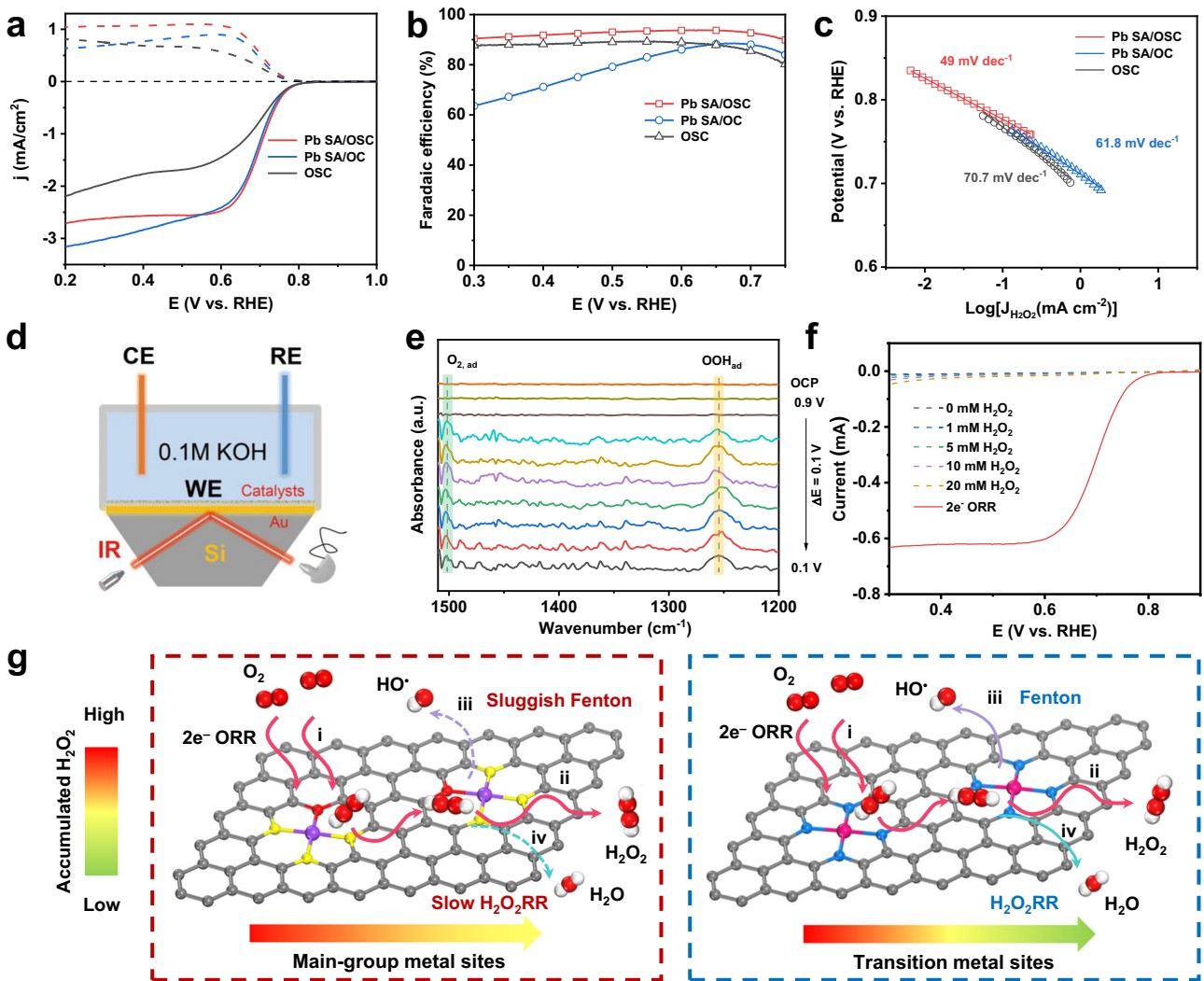

**Fig. 4 | Evaluation of two-electron ORR activity and selectivity of Pb SA/OSC using RRDE. a** Electrochemical oxygen reduction polarization curves (solid lines) along with $H_2O_2$ detection currents (dashed lines) for Pb SA/OSC, Pb SA/OC and OSC in $O_2$-saturated 0.10 M KOH electrolyte. **b** $H_2O_2$ selectivity based on the RRDE measurements. **c** Tafel slopes of catalysts derived from kinetic current for $H_2O_2$ production. **d** Schematic illustration of the experimental setup for in-situ ATR-SEIRAS measurements. WE working electrode, CE counter electrode, RE reference electrode. **e** In-situ ATR-SEIRAS spectra collected on the Pb SA/OSC catalyst in $O_2$-saturated 0.10 M KOH catholyte at various potentials ranging from 0.9 to 0.1 V *vs* RHE. **f** LSV of Pb SA/OSC in $N_2$-saturated 0.10 M KOH electrolyte containing 1 mM, 5 mM, 10 mM, or 20 mM $H_2O_2$. **g** Schematics of the electro-chemical cumulative $H_2O_2$ production on the main-group metal catalysts (left) and transition metal catalysts (right).

KOH. To study the roles of the main-group center and super-coordination structure in affecting the catalytic reactivity of Pb SA/OSC, control samples of metal-free OSC and Pb SA/OC (with O as only the coordination atom) were also prepared. Before evaluating the selectivity of $H_2O_2$, the collection coefficient of the Pt ring electrode was determined to be approximately 0.37, calibrated using the $[Fe(CN)_6]^{4-}/[Fe(CN)_6]^{3-}$ redox reaction (Supplementary Fig. 23)[47]. Figure 4a shows the linear sweep voltammetry (LSV) curves of the various catalysts, presenting the current signals for oxygen reduction (solid lines) and the corresponding partial current of $H_2O_2$ (dashed lines) recorded on the disk electrode and ring electrode, respectively. It is obvious that Pb SA/OSC exhibited a superior 2e⁻ ORR activity compared to Pb SA/OC and OSC. Pb SA/OSC exhibited the highest ring current (0.204 mA at 0.5 V vs. RHE), suggesting a higher generation of $H_2O_2$ during the ORR process. The $H_2O_2$ selectivity was calculated and plotted as a function of applied potential in Fig. 4b. In a wide potential window from 0.30 V to 0.70 V, the $H_2O_2$ selectivity of the Pb SA/OSC catalyst exceeded 90% (with a maximum of 94%), highlighting a highly selective 2e⁻ ORR pathway. In comparison, Pb SA/OC and OSC showed

much lower $H_2O_2$ selectivity and higher electron transfer number (n) values (Supplementary Fig. 24), confirming the importance of the super-coordination structure in tuning the ORR selectivity. Besides, Pb SA/OSC delivered the highest $H_2O_2$ current density ($j_{H_2O_2}$) at 0.65 V (Supplementary Fig. 25) and the most positive onset potential ($E_{onset}$) (defined as the potential at the ring current density of 0.1 mA cm⁻²), compared to the Pb SA/OC and OSC samples.

Furthermore, the Tafel slope for the Pb SA/OSC was determined to be 49 mV dec⁻¹, significantly smaller than those of Pb SA/OC (61.8 mV dec⁻¹) and OSC (70.7 mV dec⁻¹), indicating its more favorable kinetics for $H_2O_2$ generation (Fig. 4c). The electrochemically active surface areas (ECSA) were calculated by measuring the double-layer capacitances ($C_{dl}$) of the catalysts. As depicted in Supplementary Fig. 26, the Pb SA/OSC sample exhibited a larger $C_{dl}$ value and a higher availability of exposed active sites compared to the other as-prepared catalysts, making a significant contribution to its superior performance in $H_2O_2$ production. The 2e⁻ ORR activity of the Pb SA/OSC catalyst was further investigated in a neutral electrolyte (0.10 M phosphate-buffered saline, PBS, pH-7.0). As shown in Supplementary

Fig. 27, Pb SA/OSC demonstrated a high ORR current (0.56 mA at 0.1 V *vs*. RHE) and selectivity (>90%) within the potential range of 0.05–0.35 V *vs*. RHE. To confirm the role of the coordination environment in the catalytic selectivity, the ORR performance of Pb SA/OSC samples with different S contents was also studied. All samples delivered satisfactory $H_2O_2$ current density and working potential window (Supplementary Fig. 28), and the Pb SA/OSC catalyst with optimal S content exhibited the best selectivity for $2e^-$ ORR activity.

To detect the key adsorbed oxygen intermediate (*OOH) on Pb SA/OSC during electrolytic $H_2O_2$ synthesis, in-situ attenuated total reflectance surface-enhanced infrared absorption spectroscopy (ATR-SEIRAS) tests were performed (Fig. 4d). Impressively, a new absorption band at about 1254 $cm^{-1}$ emerged when applying a potential lower than 0.8 V *vs* RHE (Fig. 4e). This featured absorption band on Pb SA/OSC can be assigned to the O−O stretching vibration of *OOH, which is consistent with previous studies[48–50]. Additionally, the absorption band at around 1500 $cm^{-1}$ was attributed to the O−O stretching mode of adsorbed molecular oxygen ($O_{2,\ ad}$). The deuterated experimental results show that the vibration band of *OOH underwent a downshift to 1234 $cm^{-1}$ (Supplementary Fig. 31), indicating the involvement of hydrogen atom and confirming the origin of *OOH. Overall, the potential-dependent ATR-SEIRAS results supported the *OOH mediated $2e^-$ ORR pathway on the prepared catalysts (Supplementary Fig. 32).

To evaluate the cumulated $H_2O_2$, it is imperative to quantitatively assess the side reaction of the consecutive $H_2O_2$ reduction reaction ($H_2O_2RR$) on the same catalyst[18,26]. The electrochemical experiments for $H_2O_2RR$ were conducted in $N_2$-saturated 0.10 M KOH with varying concentrations of $H_2O_2$. As shown in Fig. 4f, the rate of $H_2O_2$ electroreduction on Pb SA/OSC exhibited only marginal increase with higher overpotential and $H_2O_2$ concentration. The current density for $H_2O_2RR$ on Pb SA/OSC was less than −0.07 mA $cm^{-2}$ when the potential exceeded 0.40 V *vs*. RHE, indicating limited activity of the main-group sites of Pb SA/OSC towards $H_2O_2RR$. Similar results were observed for main-group catalysts during the $2e^-$ ORR process[30,32]. Therefore, the Pb SA/OSC catalyst was identified as a highly promising candidates for $H_2O_2$ production owing to its minimal $H_2O_2RR$ activity and high $H_2O_2$ selectivity during ORR process. Consequently, the net rate of $H_2O_2$ production (i.e., production rate minus electroreduction rate of $H_2O_2$) on Pb SA/OSC can be maintained at a sufficiently high level as the concentration of $H_2O_2$ built up. Additionally, the potential Fenton reactions catalyzed by transition-metal sites may deplete the accumulated $H_2O_2$ and compromise the durability of the electrode[27,51]. In contrast to transition-metal sites, the main-group metal sites inherently possess lower activity towards Fenton reactions[29,52]. As shown in Supplementary Fig. 35, no characteristic signals of hydroxyl radical from Fenton reactions were observed in the EPR spectra of Pb SACs during the $2e^-$ ORR process. Hence, ORR catalysts based on main-group metal sites are expected to be an attractive option for mitigating Fenton reactions during $H_2O_2$ production. Overall, the Pb SA/OSC catalyst demonstrates significant potential in meeting the requirements for accumulative $H_2O_2$ production by suppressing $H_2O_2RR$ and Fenton reactions (Fig. 4g).

## Practical scale $H_2O_2$ electrosynthesis

Inspired by the remarkable $2e^-$ ORR selectivity and potential for $H_2O_2$ production of the Pb SA/OSC catalyst, a gas diffusion electrode (GDE) and a customized flow-cell electrolyzer were employed for practical $H_2O_2$ synthesis. As illustrated in Fig. 5a, LSV was initially performed in a flow-cell setup in 1 M KOH with manual 80% iR compensation[19,53]. As shown in Fig. 5b, the Pb SA/OSC catalyst exhibited significantly higher current density compared to the Pb SA/OC and OSC, suggesting the crucial role of the coordination environment of the Pb SA/OSC in $H_2O_2$ electrosynthesis. Bulk electrolysis experiments were then conducted

at different current densities ranging from 50 to 400 mA $cm^{-2}$ to evaluate the rates of $H_2O_2$ production. The accumulated concentration of $H_2O_2$ in the catholyte was determined using a colorimetric quantification method (Supplementary Fig. 39). Importantly, the Faradaic efficiencies (FEs) of Pb SA/OSC for $H_2O_2$ generation remained above 92% across the applied current density range (Fig. 5d). When the concentration of $H_2O_2$ builds up in the electrolyte, a high density of $H_2O_2$ leads to self-decomposition or $H_2O_2RR$[26]. Impressively, during bulk electrolysis at 200 mA $cm^{-2}$ (Fig. 5c), the concentration of the accumulated $H_2O_2$ increased almost linearly and reached 1358 mM (4 wt.%) after 2 h of electrolysis. The superior performance of the Pb SA/OSC catalyst indicated that the main-group sites with a regulated local coordination environment effectively prevented the self-decomposition or further electrochemical reduction of generated $H_2O_2$. This finding aligns with the observations obtained from the EPR measurements and $H_2O_2$ reduction reaction experiments, thereby confirming the minimal Fenton and $H_2O_2RR$ activity of the Pb SA/OSC. Furthermore, the calculations indicate that molecular $H_2O_2$ could be stabilized and accumulated at the solid−liquid interface of the Pb SA/OSC, effectively inhibiting the decomposition of $H_2O_2$. It is noteworthy that the $H_2O_2$ FEs only exhibited a slight decrease trend (97% to 91%) during constant current electrolysis at 200 mA $cm^{-2}$.

Furthermore, the Pb SA/OSC catalyst exhibited a high $H_2O_2$ production rate at different current densities ranging from 50 to 400 mA $cm^{-2}$ (Fig. 5e). At 400 mA $cm^{-2}$, the rate of $H_2O_2$ generation reached 6.9 mmol $cm^{-2}$ $h^{-1}$, ranking among the top performances of the state-of-the-art $2e^-$ ORR catalysts (Supplementary Table 7)[5,18,19,30,32,48,54–66]. To investigate the stability of Pb SA/OSC, long-term electrolysis was conducted at a constant current density of 50 mA $cm^{-2}$. As shown in Fig. 5f, a high $H_2O_2$ FE (above 93%) was maintained without noticeable degradation for at least 100 h, enabling continuous and stable production of $H_2O_2$, which is critical for industrial applications. The morphology, atomic Pb dispersion, and carbon crystallinity of the Pb SA/OSC were well-preserved after stability testing (Supplementary Figs. 44–46). This underscores the structural stability of the catalyst for the $2e^-$ ORR. Moreover, the results obtained from ICP–MS showed negligible leaching of both Pb and S (around 0.05% of Pb and 0.2% of S in Pb SA/OSC) into the bulk electrolyte after 6 h of continuous electrolysis.

The Pb SA/OSC catalyst exhibited impressive performance in electrosynthesis of $H_2O_2$. To further evaluated its practical applications, we coupled the obtained $H_2O_2$ solution with biomass-derived carbohydrate conversion for the production of formic acid (HCOOH). Previous studies have reported that biomass-derived carbohydrates can be transformed to formic acid, an important $H_2$ carrier, through hydrothermal oxidation in the presence of $H_2O_2$ and alkali[67,68]. Taking glucose to HCOOH conversion as a representative example, a high HCOOH yield was obtained using alkali as a catalyst and $H_2O_2$ as an oxidant. The alkali played a crucial role in facilitating selective oxidation and inhibiting overoxidation of the formed HCOOH (Supplementary Fig. 47). In this regard, the $H_2O_2$ solution obtained by electrosynthesis provided an ideal platform for biomass-derived carbohydrate conversion, as it supplied both alkali (KOH) and $H_2O_2$ simultaneously. The accumulated $H_2O_2$ solution (1358 mM) obtained by bulk electrolysis (200 mA $cm^{-2}$ for 2 h) was used for the hydrothermal oxidation of glucose at different temperatures. For comparison, commercial $H_2O_2$ with an optimized amount of KOH was used under the same conditions for the above reactions. As shown in Fig. 5g, both the electrochemically produced $H_2O_2$ solution and the commercial $H_2O_2$ delivered extraordinary yields of HCOOH (over 90%) at temperatures of 150 and 180 °C. Additionally, a slight decrease in the yield of HCOOH was observed at a lower temperature (120 °C). The electrocatalytically produced $H_2O_2$ solution can be directly applied for biomass-derived carbohydrate conversion without further treatment.

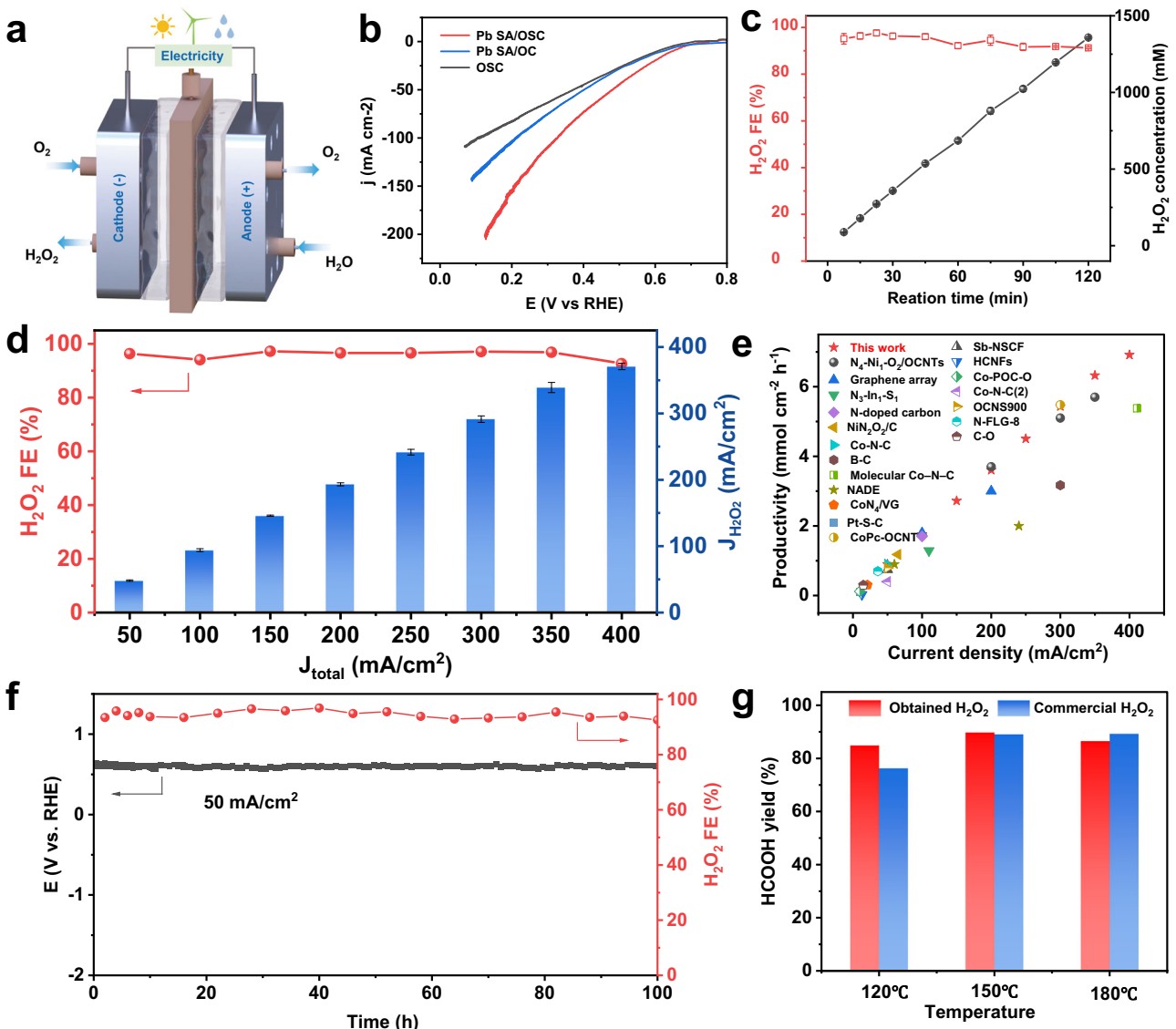

**Fig. 5 | Electrosynthesis performance of H₂O₂ in a flow-cell electrolyzer based on the GDE. a** Schematic illustration of the flow-cell device. **b** LSV curves for Pb SA/OSC, Pb SA/OC, and OSC. The curves are manually 80% iR-compensated (R = 0.4 ± 0.1 Ω). **c** The accumulated H₂O₂ concentrations and Faradaic efficiencies of Pb SA/OSC at 200 mA cm⁻². **d** H₂O₂ Faradaic efficiencies and $J_{H2O2}$ of Pb SA/OSC at different current densities. Error bars in (**c**, **d**) correspond to the standard deviation of three independent measurements. **e** H₂O₂ production rate of Pb SA/OSC compared to the reported state-of-the-art electrocatalysts. **f** Stability test and H₂O₂ selectivity of the Pb SA/OSC catalyst under 50 mA cm⁻², with the electrolyte refreshed every 6 h during the test. **g** Yield of formic acid from the oxidation of glucose by the produced and commercial H₂O₂ at different temperatures.

## Discussion

In summary, we developed a main-group single-atom electrocatalyst featuring S, O super-coordinated Pb sites anchored in carbon frameworks through a carbon dot-assisted pyrolysis approach. Through rational engineering of the coordination environments of the main-group sites, the Pb SA/OSC catalysts exhibited remarkable performances in the 2e⁻ ORR toward H₂O₂ with a high selectivity of up to 94%. Furthermore, industrial current densities up to 400 mA cm⁻² with 92–97% FE for H₂O₂ production were achieved at an impressive rate of 6.9 mmol cm⁻² h⁻¹ in a GDE-based flow-cell electrolyzer in an alkaline medium, surpassing the performance of most of the state-of-the-art 2e⁻ ORR catalysts. The Pb SA/OSC catalyst maintained adequate stability without an obvious decrease in H₂O₂ production and Faradaic efficiency during 100-h continuous electrolysis. Notably, due to the inert activity of the main-group Pb SACs for the H₂O₂RR and Fenton-like reactions, the concentration of the accumulated H₂O₂ solution

reached 1358 mM with high FE during bulk electrolysis. Moreover, the obtained H₂O₂ solution was successfully utilized for the hydrothermal oxidation of renewable biomass-derived glucose, delivering an extraordinary yield of HCOOH (over 90%). Combined experimental and theoretical studies suggest that the superior catalytic performance of Pb SA/OSC in the 2e⁻ ORR can be attributed to the S, O super-coordination regulated microenvironments of the Pb-S₄O₂ moieties. This work presents a promising avenue for the design of 2e⁻ ORR catalysts to address the production-depletion dilemma in H₂O₂ electrosynthesis and could open the door for green production of formic acid from renewable biomass.

## Methods
### Chemicals
Acetaldehyde (40% water solution), sodium hydroxide, potassium hydroxide, hydrochloric acid, lead (II) acetate trihydrate (≥99.5%),

glucose and anhydrous ethanol were supplied by Sinopharm Chemical Reagent Co., China. 2-Mercaptopropionic acid (98%) and titanium (IV) oxysulfate-sulfuric acid hydrate (93%) were purchased from Macklin Co., China. Nafion solution (5 wt.%) was provided by Sigma-Aldrich Co., USA. Carbon black (Vulcan XC-72) was bought from The Cabot Co., USA. All chemicals and reagents were directly used as received without further purification.

## Preparation of SCDs and OCDs

The sulfur-doped carbon dots (SCDs) were synthesized by using a high-efficiency aldol condensation method at room temperature[69]. In a typical preparation procedure, 5 mL of 2-mercaptopropionic acid was dissolved in 40 mL of acetaldehyde. Then, 15 g of NaOH was slowly added and stirred for 2.5 h. The product obtained after the completion of the reaction was sonicated with a certain amount of deionized water for 30 min. An aqueous solution of the crude product was centrifuged (10,000 rpm/min for 10 min to remove large or agglomerated particles. The supernatant was collected and transferred into a dialysis bag (MWCO: 3500 Da) to dialyze for two days.

The SCDs samples with different S contents were prepared by using a similar procedure except for changing the amount of 2-mercaptopropionic acid to 0, 2.5, and 7.5 mL. They are denoted as OCDs, SCDs-2.5 and SCDs-7.5, respectively.

## Synthesis of Pb SA/OSC

In a typical synthesis of the Pb SA/OSC catalyst, first lead precursor solution was prepared by dissolving $Pb(CH_3COO)_2$ in $H_2O$ to achieve a 24 mM $Pb(CH_3COO)_2$ stock solution. Subsequently, 300 mg of purified SCDs was dispersed in 50 mL of $H_2O$ under sonication for 10 min to obtain the SCDs solution. Then, 1 mL of $Pb(CH_3COO)_2$ solution and 300 mg of Vulcan XC-72 in ethanol (10 mL) were added to the above solution, followed by sonication for 15 min. Afterward, the above-mixed solution was dried first by rotary evaporation and then in a vacuum oven at 80 °C. The resulting black powder was heated to 300 °C (heating rate, 5 °C min$^{-1}$) for 2 h in a nitrogen flow. After thorough washing with water and ethanol, the dried powders were subjected to high-temperature annealing at 550 °C (heating rate, 5 °C min$^{-1}$) for 2 h in a nitrogen flow. After being naturally cooled to room temperature, the Pb SA/OSC catalyst was obtained.

For the Pb SA/OSC-2.5 and Pb SA/OSC-7.5 samples, the synthesis process is the same as that of Pb SA/OSC, but the S-doped carbon dots are SCDs-2.5 and SCDs-7.5, respectively. Moreover, the OSC sample was prepared by the same procedure as described for Pb SA/OSC but without any Pb precursor addition.

## Synthesis of Pb SA/OC

For the Pb SA/OC catalyst preparation, 300 mg of purified OCDs was dispersed in 50 mL of $H_2O$ under sonication for 10 min to obtain the OCDs solution. Then, 1 mL of $Pb(CH_3COO)_2$ solution and 300 mg of Vulcan XC-72 in ethanol (10 mL) were added to the above solution, followed by sonication for 15 min. Afterward, the above-mixed solution was dried first by rotary evaporation and then in a vacuum oven at 80 °C. The resulting black powder was heated to 300 °C (heating rate, 5 °C min$^{-1}$) for 2 h in a nitrogen flow. After thorough washing with water and ethanol, the dried powders were subjected to high-temperature annealing at 550 °C (heating rate, 5 °C min$^{-1}$) for 2 h in a nitrogen flow. After being naturally cooled to room temperature, Pb SA/OC was obtained.

## Characterizations

The morphologies of the electrocatalysts were characterized by transmission electron microscopy (TEM) using a microscope (7700, Hitachi Co., Japan) with an accelerating voltage of 100 kV. Aberration-corrected high-angle annular dark-field scanning transmission electron microscopy (HAADF-STEM) images and element mapping were obtained on a STEM instrument (JEM-ARM200CF, JEOL Co., Japan)

incorporated with double spherical aberration correctors operated at 200 kV. X-ray diffraction (XRD) measurements were recorded on a Rigaku Miniflex-600 diffractometer using Cu Kα radiation (λ = 0.15406 nm) with a step size of 0.02° and a counting time of 0.5 s. X-ray photoelectron spectroscopy (XPS) was collected on a scanning X-ray microprobe (PHI 5000 Versasa, ULAC-PHI Inc., USA) using Al Ka radiation and the C1s peak at 284.8 eV as an internal standard. Raman spectra were taken using a Raman spectroscopy instrument (LabRAM HR Evolution, Horiba Co., Japan) equipped with a green laser emitting at 532 nm. Elemental analysis of Pb in the solid samples analyzed with inductively coupled plasma–mass spectrometry (ICP–MS).

Soft X-ray absorption spectra (Soft-XAS) were carried out at the Catalysis and Surface Science endstation at the BL11U Beamline and Photoemission Endstation at the BL10B beamline in the National Synchrotron Radiation Laboratory (NSRL) in Hefei, China. X-ray absorption fine structure (XAFS) spectroscopy measurements were performed at the Pb L$_3$-edge. Data were collected at the BL14W1 beamline of the Shanghai Synchrotron Radiation Facility (SSRF), China. The storage ring was operated at 3.5 GeV with a maximum electron current of 250 mA. Pb foil and PbS were used as references. The Pb L$_3$-edge X-ray absorption near-edge spectroscopy (XANES) data were recorded in fluorescence mode. The hard X-ray was monochromatized with Si (111) double-crystals. The acquired EXAFS data were extracted and processed according to standard procedures using the ATHENA module implemented in the IFEFFIT software package. The k$^2$-weighted EXAFS spectra were obtained by subtracting the post-edge background from the overall absorption and then normalizing with respect to the edge-jump step. Subsequently, k$^2$-weighted χ(k) data in the k-space ranging from 2.5–11.2 Å$^{-1}$ were Fourier transformed to real (R) space using hanning windows (dK = 1.0 Å$^{-1}$) to separate the EXAFS contributions from different coordination shells. In situ attenuated total reflection surface-enhanced infrared absorption spectra (ATR-SEIRAS) measurements were performed on a Nicolet™ iS50 FT-IR spectrometer (Thermo Scientific Co., USA) with an MCT detector.

## Electrochemical measurements

The electrochemical experiments were performed in a standard three electrode system controlled by a CHI 760e workstation (CH Instruments Co., China). A rotating ring disk electrode (RRDE, Pine Research Instrumentation, USA) consisting of a glassy carbon electrode (disk area: 0.2475 cm$^2$) and a platinum ring electrode (ring area: 0.1866 cm$^2$) was used as the working electrode, with a theoretical collection efficiency of 37%. An Ag/AgCl electrode (saturated KCl solution) and a Pt wire were used as the reference and counter electrodes, respectively. All the recorded potentials were normalized by converting to reversible hydrogen electrode (RHE) according to the equation $E_{RHE} = E_{Ag/AgCl} + 0.197 + 0.059$ pH. The potential versus RHE was adopted unless otherwise specified.

To prepare the working electrode, 4 mg of the catalysts and 40 μL of Nafion solution (5 wt.%) were mixed with 960 μL of 1:1 (vol/vol) water/ethanol mixed solution and then dispersed by ultra-sonication for 30 min until a homogeneous catalyst ink was obtained. After polishing the RRDE mechanically with an alumina suspension, 7.5 μL of catalyst ink was dropped on the disk electrode and dried to form a uniform thin film at room temperature. All electrochemical measurements were carried out in $O_2$ or $N_2$-saturated 0.10 M KOH and 0.10 M PBS aqueous solution. Prior to the ORR measurement, the pre-activation process by scanning CV scans at a rate of 50 mV s$^{-1}$ from 0.05 to 1.0 V was performed on the RRDE until a stable state was reached. Then, the ORR polarization curves were acquired from LSV measurements under $O_2$-saturated conditions between 0.05 and 1.0 V at a scan rate of 10 mV s$^{-1}$ at 1600 rpm. While the Pt ring potential was held at 1.2 V to quantify the amount of generated $H_2O_2$ at the disc electrode. Especially, the Pt ring electrode was electrochemically cleaned through scanning CV curves for 20 cycles

(potential range: 0.3–0.05 V, scanning rate: 100 mV/s) on the Pt ring before LSV measurement.

The collection efficiency (N) on the RRDE electrode was calibrated in 0.10 M KOH + 10 mM $K_3[Fe(CN)_6]$ electrolyte at different rotation rates. Thus, the measured collection efficiency was determined to be 37.08%, which is reasonably close to the theoretical value. The $H_2O_2$ reduction reaction ($H_2O_2RR$) was studied by conducting LSV measurements of the catalyst-coated disk electrode in $N_2$-saturated 0.1 M KOH solution containing 1 mM, 5 mM, 10 mM, or 20 mM $H_2O_2$ at 1600 rpm.

The selectivity of hydrogen peroxide ($H_2O_2$ %) and electron transfer number (n) were calculated with the following equation:

$$H_2O_2\% = 200 \frac{I_R/N}{|I_D| + I_R/N} \tag{1}$$

$$n = 4 \frac{|I_D|}{|I_D| + I_R/N} \tag{2}$$

where $I_R$ is the ring current, $I_D$ is the disk current and N is the determined collection efficiency.

The kinetic current density ($j_k$) was calculated using the Koutecký–Levich equation:

$$\frac{1}{j} = \frac{1}{j_l} + \frac{1}{j_k} \tag{3}$$

where $j$ is the measured current density and $j_l$ is the diffusion-limited current density. The $j$ value is acquired by dividing the ring current by the collection efficiency (N) and the disk electrode area. The $j_l$ value was taken from the highest value in the j plot measured over the entire potential range investigated (0.05–1.0 V).

Tafel slopes were calculated according to the Tafel equation:

$$\eta = a + blog(j_k) \tag{4}$$

where $a$ is the constant, $\eta$ is the applied potential, $b$ is the Tafel slope, and $j_k$ is the kinetic current density for $H_2O_2$ production.

Electrochemical capacitance measurements were conducted from double-layer charging curves using the CVs at the non-Faradic potential window in 0.10 M KOH solution. These samples were tested with scan rates ranging from 10 to 100 mV s$^{-1}$. The plots of current density (at 1.05 V) as a function of the scan rate showed the double-layer capacitance by the slopes.

## Bulk electrosynthesis of $H_2O_2$

The cathode was fabricated by spray-coating the as-synthesized Pb SA/OSC catalyst on a gas diffusion electrode (GDE, YLS30, Suzhou Sinero Technology Co., China). In a typical process, 8 mg of catalyst was mixed with 80 μL of 5 wt% Nafion solution and 4 ml of 1:1 (vol/vol) ethanol/water mixed solvent. The above solution was sonicated for at least 60 min to form a homogeneous ink. The well-dispersed ink was then airbrushed onto a GDE (2.5 × 2.5 cm$^2$) with a loading amount of 1.0 mg cm$^{-2}$ (Supplementary Fig. 41). The gas diffusion electrode (GDE) was dried in a vacuum chamber overnight and cut to a size of 2.0 × 2.0 cm$^2$. A Ti-mesh electrode coated with $IrO_2$ and an Ag/AgCl electrode (saturated KCl) were employed as the anode and reference electrode, respectively. The anode was prepared by depositing $IrO_2$ on a titanium support by a dip coating and thermal decomposition method[70]. Briefly, the Ti mesh was cut to a size of 2.0 × 2.0 cm$^2$ area and washed with acetone and DI water, and then ultrasonicated in a 6 M HCl aqueous solution heated to 80 °C for 45 min before dip coating. The coating stock solution consisted of 30 mg of $IrCl_3 \cdot xH_2O$ (Alfa Aesar) dissolved in 10 mL of an isopropanol solution with 10% concentrated HCl. The Ti mesh was dipped into the $IrCl_3$ solution, followed by drying in a vacuum oven at 100 °C for 10 min before calcination in a

furnace at 500 °C for 10 min. The dipping and calcination process was repeated until a 2 mg cm$^{-2}$ loading was achieved.

Practical $H_2O_2$ production was carried out using a gas-tight flow cell system with an $O_2$ gas compartment and two liquid compartments. The catholyte and anolyte chambers were separated by a cation exchange membrane (Nafion 117, Fuel Cell Store Co., USA). The electrolyzer cell consisted of two titanium backplates with a 4.0 cm$^2$ serpentine flow field. During the electrocatalytic tests, the electrolyte (1 M KOH) was recycled at a constant flow rate of approximately 10 mL min$^{-1}$ with a peristaltic pump, and the $O_2$ gas feed was maintained at 50 mL min$^{-1}$ flowing through the cathode using a flow controller. The constant currents were provided by a workstation (PMC-1000, PARSTAT Inc., USA). For the test of the $H_2O_2$ production rate, the cathode current density was increased from 50 to 400 mA cm$^{-2}$. The 1 M KOH electrolyte was refreshed periodically every 6 h in the stable $H_2O_2$ production up to 100 h. The $H_2O_2$ concentration was quantitatively determined by the titanium sulfonate $Ti(SO_4)_2$ titration method ($TiOSO_4 + H_2O_2 + H_2SO_4 = H_2[Ti(O_2)(SO_4)_2] + H_2O$). After the addition of $H_2O_2$, a yellow solution of $H_2[Ti(O_2)(SO_4)_2]$ was produced. A linear calibration relationship between the $H_2O_2$ concentration and the $H_2[Ti(O_2)(SO_4)_2]$ absorbance peak at 408 nm was acquired using UV–vis spectroscopy. Here, 2 wt% $TiOSO_4$ in 2 M $H_2SO_4$ was prepared by stirring overnight until a transparent solution was formed, which was used to spectrophotometrically quantify the high concentration of $H_2O_2$ produced in the flow cell.

Electrochemical impedance spectroscopy (EIS) was conducted at 0.75 V (vs. RHE) from 100,000 to 0.1 Hz to determine the uncompensated resistance in a high-frequency range for iR-correction in the electrochemical cell.

The faradaic efficiency of $H_2O_2$ production ($FE_{H2O2}$) was determined by the following equation:

$$H_2O_2(FE,\%) = 2 \times \frac{C_{H2O2} \cdot V \cdot F}{I \cdot t} \times 100 \tag{5}$$

where $C_{H2O2}$ is the produced $H_2O_2$ concentration (mol L$^{-1}$) in the electrolyte, $V$ refers to the volume of electrolyte (L), F indicates the Faraday constant (96485 C mol$^{-1}$), $I$ and $t$ are the operating current (A) and test time (s), respectively.

## Glucose oxidation for formic acid and product analysis

The selective oxidation of glucose was studied by batch experiments. In a typical experiment, 20 mL of the alkaline $H_2O_2$ solution in the flow cell was cycled for 2 h at 200 mA cm$^{-2}$. Then, 4 mL of the above alkaline $H_2O_2$ solution and 70 mg of glucose were introduced into the reactor. Then, the reactor was transferred to an oven preheated to the required temperature (120, 150 or 180 °C). After 15 min, the reactor was removed from the oven and immersed in a cold-water bath. By comparison, the same amount of commercial $H_2O_2$ with the optimized concentration of KOH (2.5 M) were used for the above reactions under the same conditions. After the reaction, the pH of the solution was adjusted to 3 with HCl for high-performance liquid chromatography (HPLC) analysis (LC-16, Shimadzu Co., Japan). HPLC analyses were performed on an Amine® HPX-87H column with a refractive index detector.

## Computational methods

Density functional theory (DFT) calculations were performed by Vienna Ab initio Simulation Package (VASP) with the projector augment wave (PAW) method[71–73]. The electron exchange and correlation energy were treated within the generalized gradient approximation with the Perdew-Burke-Ernzerhof (GGA-PBE)[74]. Spin polarization was taken into account in all calculations. The k-points sampling for Brillioun zone was set as 1 × 1 × 1. The convergence criteria for geometry (self-consistent field) were set to 10$^{-4}$ eV for energy and 0.02 eV/Å for forces. The cutoff energy for the kinetic energy of the plane waves was

400 eV. The electronic configuration of lead is $[Xe]6s^24f^{14}5d^{10}6p^2$ and a $5d^{10}6s^26p^2$ valence electron potential was considered in the calculations. Partial occupancies were determined using a Gaussian smearing scheme with a smearing width of 0.05 eV. All geometries were fully relaxed to the ground state. The structures of the $PbS_4O_2$-water interface were sampled through the random placement of 36 water molecules in the water-slab region followed by local optimization started with two sets of configurations. To describe the electrostatics and dispersion at the interface, the polarizable continuum model implemented in VASPsol with relative permittivity of 78.4 as water was employed[75]. The van der Waals interactions were considered using the empirical DFT-D3 method proposed by Grimme et al.[76]. The ground state structures of adsorbed *OOH, *O and *OH were determined by searching the lowest energy one among all the possible configurations on possible active sites.

Gibbs free energy change ($\Delta G$) was calculated as $\Delta G = \Delta E + \Delta ZPE - T\Delta S + \Delta G_U + \Delta G_{pH}$, where $\Delta E$ is the electronic energy difference directly obtained from DFT calculations, $\Delta ZPE$ is the zero-point energy difference, $T$ is the room temperature (298.15 K) and $\Delta S$ is the entropy change. $\Delta G_U = -eU$ where $U$ is potential at the electrode, e is charge transferred and $\Delta G_{pH} = k_BT \times \ln10 \times pH$ where $k_B$ is the Boltzmann constant and T = 300 K. $O_2$ reduction to $H_2O_2$ follows a two-electron reaction mechanism as below:

$$\Delta G_1 : O_2(g) + * + H^+ + e^- \rightarrow *OOH \tag{6}$$

$$\Delta G_2 : *OOH + H^+ + e^- \rightarrow H_2O_2(l) \tag{7}$$

where asterisk (*) denotes the Pb site of catalyst and *OOH denotes the intermediate adsorbed on the catalyst surface. The overpotential that evaluates the performance of the $2e^-$ ORR is obtained as $\eta_{ORR} = \max\{\Delta G_1, \Delta G_2\}/e + 0.7$, where 0.7 represents the equilibrium potential. The limiting potential is the potential at which the reaction step becomes exergonic, $U_{L(O2/H2O2)} = -\max\{\Delta G_1, \Delta G_2\}/e$.

We assume pH = 0 for acidic medium. The associative four-electron reaction is composed of the following elementary steps:

$$\Delta G_3 : O_2(g) + * + H^+ + e^- \rightarrow *OOH \tag{8}$$

$$\Delta G_4 : *OOH + H^+ + e^- \rightarrow *O + H_2O(l) \tag{9}$$

$$\Delta G_5 : *O + H^+ + e^- \rightarrow *OH \tag{10}$$

$$\Delta G_6 : *OH + H^+ + e^- \rightarrow H_2O(l) + * \tag{11}$$

The overpotential of the $4e^-$ ORR is obtained as $\eta_{ORR} = \max\{\Delta G_3, \Delta G_4, \Delta G_5, \Delta G_6\}/e + 1.23$, where 1.23 represents the equilibrium potential. The limiting potential is determined by $U_{L(O2/H2O)} = -\max\{\Delta G_3, \Delta G_4, \Delta G_5, \Delta G_6\}/e$. Adsorption free energies of any reaction intermediate i under standard conditions are conveniently calculated as $G_{ad}(i) = E_{ad}(i) + G_{corr}(i)$, where $E_{ad}(i)$ is the adsorption energy and $G_{corr}(i)$ is free energy contributions to the free adsorbate. The values for $G_{corr}(i)$ for adsorbates at the Pb site of $PbS_4O_2$ catalysts are provided in Supplementary Table 5. All adsorption energies are then consistently referenced relative to the stable molecules of $H_2O(l)$ and $H_2(g)$ in Supplementary Table 6, together with experimental energy change of reduction $O_2$ to $H_2O$ (4.92 eV) to construct the free energy diagram. The free energies of $O_2$, *OOH and $H_2O_2$ at a given potential $U$ relative to normal hydrogen electrode (NHE)[77–79] are defined as follows:

$$\Delta G(O_2) = 4.92 - 2eU \tag{12}$$

$$\Delta G(*OOH) = G(*OOH) - 3.52 - eU \tag{13}$$

$$\Delta G(H_2O_2) = 3.52 \tag{14}$$

We assume pH = 14 for alkaline medium. The corresponding associative four-electron reaction is composed of the following elementary steps:[80]

$$O_2(g) + * + H_2O + e^- \rightarrow *OOH + OH^- \tag{15}$$

$$*OOH + e^- \rightarrow *O + OH^- \tag{16}$$

$$*O + H_2O + e^- \rightarrow *OH + OH^- \tag{17}$$

$$*OH + e^- \rightarrow * + OH^- \tag{18}$$

## Data availability

The data supporting the findings of this study are available within the article and its Supplementary Information files. All other relevant source data are available from the responding author upon request. Source data of the article are provided with this paper. Source data are provided with this paper.

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

## Acknowledgements

This work was supported by the National Key Research and Development Program of China (2021YFA1501003 to Y.W.), the National Natural Science Foundation of China (52100195 to X.Z., 51821006 to H.Q.Y., 52027815 to H.Q.Y., 52192684 to H.Q.Y., 92261105 to Y.W. and 22221003 to Y.W.), the Anhui Provincial Natural Science Foundation (2108085QB70 and 2108085UD06 to Y.W.), the Key Technologies R&D Program of Anhui Province (2022a05020053 to Y.W.), the Students' Innovation and Entrepreneurship Foundation of University of Science and Technology of China (CY2022G42 to S.L.X.) and the Collaborative Innovation Program of Hefei Science Center, CAS (2021HSC-CIP002 to Y.W.). We thank the photoemission endstations BL1W1B in Beijing Synchrotron Radiation Facility (BSRF), BL14W1 in Shanghai Synchrotron Radiation Facility (SSRF), BL10B and BL11U in National Synchrotron Radiation Laboratory (NSRL) for the help in characterizations.

## Author contributions

H.Q.Y. and Y.E.W. conceived and designed the study and supervised the project. X.Z. carried out the synthesis, characterization, and electrochemical tests. Y.M. and J.J.C. conducted the theoretical studies. C.M.Z. and M.K.K. helped to synthesize the catalysts. C.C. and S.L.X. helped with XANES and EXAFS measurements and discussion. X.Z., Y.M., H.Q.Y., and Y.E.W. analysed the results and co-wrote the paper. All authors contributed to discussion of the results and the manuscript.

## Competing interests

The authors declare no competing interests.
