## [Peer Review File · Nature Communications]

REVIEWER COMMENTS

Reviewer #1 (Remarks to the Author):

In this manuscript, the authors report a method employing carbon-dot-assisted pyrolysis to prepare main-group Pb SACs (Pb SA/OSC) for enhanced 2e- ORR performance in alkaline media. The topic is interesting. However, according to the DFT and experimental results, there are still some problems in this manuscript. Given these following comments, a minor revision is needed.

- 1) For the RRDE test, the ORR performance of Pb SA/OSC in neutral medium (0.10 M PBS) was determined. Why did the authors not add tests utilizing gas diffusion electrodes under the same conditions?
- 2) For the flow cell test, the authors used carbon paper as a base to form a gas diffusion electrode. However, many literatures have reported carbon-based materials showed good performance towards 2e- ORR. The contribution of the carbon paper to the ORR performance needs to be clarified.
- 3) The authors mentioned “The superior performance of the Pb SA/OSC catalyst indicated that main-group sites with a regulated local coordination environment effectively prevented the self-decomposition or further electrochemical reduction of generated H₂O₂”. However, the experimental and theoretical data do not support this conclusion. The authors need discuss more in-depth.
- 4) For stability testing, the authors did not give sufficient characterizations to prove the stability of catalyst after 100 h of electrolysis.

Reviewer #2 (Remarks to the Author):

This study shows how Pb single sites coordinated with sulfur can increase the H₂O₂ selectivity. The study is unique in their approach with stitching together fundamental first principles studies, lab scale tests, a battery of characterisation studies, industrial scale tests, and an application of using the H₂O₂ for an oxidation reaction. The amalgamation of these factors create a unique study that is potentially of interest to Nature Communications. But, before that, a few technical points need to be revised, the manuscript in its current form is not suited for Nature Communications. Revising these points is critical to justify overall conclusions. The primary concerns are with explaining the rationale for selecting the DFT structures, not considering oxidized sulfur sites when evidence suggests otherwise, and not reporting the extent of sulfur leaching.

1. Figure 1 shows the structures used for DFT simulations. Why were oxidized sites not considered, when such oxidation of sulfur is expected and even seen experimentally (XPS studies, C-SO_x peaks)?

2. What was the rationale for selecting these structures?
3. Can the free energy diagram be replotted at the equilibrium potential? Th
4. Does the sulfur leach out? Pb leaching has been reported but what about other elements? Can pre/post characterisations be performed to confirm this point?
5. Figure 4b: The three sites, Pb SA/OSC, Pb SA/OC and OSC all show faradaic efficiencies in the same range. What are the error bars on the measurements? Are DFT calculations qualitatively accurate, especially using a simple limiting potential analysis, to differentiate the relative performance of these sites when their farafaic efficiencies vary by less than 10% at 0.6-0.7 V?
6. In the SI, the authors state that the adsorption configurations of O*/OH*/OOH* were determined after searching across all possible configurations. Please report the configurations scanned and the adsorption energies determined in the SI, in addition to reporting the most stable structures.
7. The captions in the SI need more detail so that they can be stand-alone with the figure. Please expand on the captions in the SI describing key aspects of the figure. This change will improve readability.
8. What are the vibrational frequencies used to determine the ZPE? Were solvation effects included? Please report these aspects for greater reproducibility.
9. Line 188, Main text etc: Please round all adsorption energies calculated using DFT to two decimal places keeping in mind the general numerical accuracies of these simulations.

Reviewer #3 (Remarks to the Author):

This work demonstrates the synthesis of a Pb SA/OSC catalyst, which exhibited relatively promising H₂O₂ electroynthesis activity and selectivity with remarkable stability (100h) in an alkaline electrolyte. The Pb SA/OSC catalyst was applied to a more practical gas diffusion electrode-based reactor and a high current density of 400 mA cm⁻² was achieved.

This catalyst is interesting and has not been reported for 2-e ORR so far. However, a few technical issues must be fully addressed to meet the high standard of Nature Communications. Detailed comments are listed below:

1. Some recent advancements in theoretical understanding and material innovation have led to the development of a series of efficient main group metal single-atom catalysts for the electrosynthesis of H₂O₂ in alkaline media (Angew. Chem. Int. Ed. 2022, 134, e202117347). In comparison with the activity, the Pb SA/OSC catalyst showed excellent stability in the flow cell. More theoretical discussion is needed to explain this advantage.
2. In Figure 1c, the DFT calculations on the Gibbs free energy should be reconducted to consider the related species in alkaline media (OH⁻ and HO₂⁻) instead of those in acid (H₂O and H₂O₂). The authors could refer to some references, such as Phys. Chem. Chem. Phys., 2013, 15, 148-153.
3. Operando FTIR Spectroscopy of Pb SA/OC and OSC should be offered to prove the adsorption of O₂ and OOH on Pb SAs rather than S and O. Also, isotope experiment is required to confirm the origin of OOH.
4. Thiocyanide (SCN⁻) poisoning experiment is needed to probe the active site of Pb SA/OSC.
5. To demonstrate the high stability of the Pb SA/OSC catalyst, some post characterizations are required.
6. The structural information was gained from the EXAFS fitting analysis, and the fitting errors must be provided.

Response to Reviewer 1's comments

In this manuscript, the authors report a method employing carbon-dot-assisted pyrolysis to prepare main-group Pb SACs (Pb SA/OSC) for enhanced 2e⁻ ORR performance in alkaline media. The topic is interesting. However, according to the DFT and experimental results, there are still some problems in this manuscript. Given these following comments, a minor revision is needed.

Reply: We express our profound gratitude for the reviewer's commendation and strong support of our work's suitability for publication, as well as the crucial suggestions that have enhanced the quality of our manuscript.

1. For the RRDE test, the ORR performance of Pb SA/OSC in neutral medium (0.10 M PBS) was determined. Why did the authors not add tests utilizing gas diffusion electrodes under the same conditions?

Reply #1.1: Gas diffusion electrodes were utilized to evaluate the 2e⁻ ORR performance of Pb SA/OSC in a neutral medium. To ensure an adequate electrolyte concentration for high current densities, 1 M Na₂SO₄ solution was used in the gas diffusion electrode tests. As shown in Figure R1, the Faradaic efficiencies (FEs) of Pb SA/OSC for H₂O₂ generation in the neutral medium remained above 91.9% across the applied current densities ranging from 50 to 300 mA/cm². It can be concluded that Pb SA/OSC also demonstrated exceptional efficacy in producing H₂O₂ via gas diffusion electrodes within a neutral medium.

The data depicted in Figure R1 has been incorporated into the revised Supplementary information (Figure S43).

Figure R1: H₂O₂ Faradaic efficiencies and J_{H₂O₂} of Pb SA/OSC using gas diffusion electrodes in a neutral medium.

2. For the flow cell test, the authors used carbon paper as a base to form a gas diffusion electrode. However, many literatures have reported carbon-based materials showed good performance towards 2e⁻ ORR. The contribution of the carbon paper to the ORR performance needs to be clarified.

Reply #1.2: We agree the reviewer’s perspective that carbon-based materials have demonstrated good performance towards $2e^-$ ORR, as corroborated by numerous literatures. Typically, a GDE comprises of two distinct layers: a carbon fiber layer and a carbon black layer infused with polytetrafluoroethylene. The carbon black layer, situated between the carbon fiber layer and the catalyst layer, has the potential to exhibit $2e^-$ ORR activity within a flow-cell electrolyzer.

To clarify the contribution of the carbon paper to the ORR performance, a blank GDE (YLS30, Suzhou Sinero Technology Co., China) devoid of the catalyst layer was employed for H_2O_2 production at a current density of 50 mA/cm^2 (Figure R2). In the initial 60 seconds of electrolysis, the blank GDE necessitated a notably elevated overpotential to activate the carbon black nanoparticles, thereby reaching the desired current density. In comparison with the Pb SA/OSC catalyst, the blank GDE exhibited a substantially higher overpotential (exceeding 150 mV) to sustain a consistent current density throughout the electrolysis, despite both the blank GDE and Pb SA/OSC showcasing notable H_2O_2 Faradaic efficiencies.

Figure R2: (a, b) Chronopotentiometry tests and H_2O_2 Faradaic efficiencies of the blank GDE and Pb SA/OSC at a current density of 50 mA/cm^2 in a flow-cell electrolyzer.

Subsequently, we carried out bulk electrolysis for H_2O_2 production using the blank GDE at 200 mA/cm^2 . As shown in Figure R3a, the H_2O_2 Faradaic efficiencies of the blank GDE were lower than those of the Pb SA/OSC sample and exhibited a decline over the course of electrolysis. It has been documented that the oxygen content at the GDE surface undergoes a substantial increase under cathodic conditions, rendering the GDE hydrophilic (*Science*, 2018, 360, 783–787; *Energy Environ. Sci.*, 2021, 14, 1959–2008). It is plausible that the electrolysis for H_2O_2 production led to the grafting of numerous oxygen groups onto the carbon black layer of the blank GDE. Consequently, the GDE lost its hydrophobicity and became penetrated by the aqueous electrolyte, resulting in a decline in H_2O_2 Faradaic efficiency and structural damage to the GDE. Subsequent observations revealed the destruction and separation of the GDE into distinct carbon fiber and carbon black layers after 2 h of electrolysis at 200 mA/cm^2 (Figure R3b).

The aforementioned analysis suggests that the carbon paper without the catalyst layer necessitated a substantial overpotential to enable the carbon black layer to exhibit

$2e^-$ ORR activity, likely resulting in structural impairment to the GDE. Therefore, the contribution of the carbon paper to the ORR performance appeared to be negligible.

Figure R3: (a) H_2O_2 Faradaic efficiencies of the blank GDE and Pb SA/OSC at 200 mA/cm^2 in a 1 M KOH solution. (b) Digital image of the blank GDE after 2 h of electrolysis.

3. The authors mentioned “The superior performance of the Pb SA/OSC catalyst indicated that main-group sites with a regulated local coordination environment effectively prevented the self-decomposition or further electrochemical reduction of generated H_2O_2 ”. However, the experimental and theoretical data do not support this conclusion. The authors need discuss more in-depth.

Reply #1.3: Main-group sites are characterized by fully occupied d -orbitals, which lack of a combination of both empty and occupied host orbitals. Generally, these main-group sites exhibit inert activity towards electron transfer reactions, which, in turn, has the potential to mitigate the risks associated with the self-decomposition or electrochemical reduction of H_2O_2 .

Firstly, it is necessary to mention that main-group catalysts have been established as ORR catalysts with enhanced durability by mitigating transition metal-induced Fenton reactions. This has been substantiated by a collection of excellent works (*Nat. Mater.*, 2020, 19, 1215-1223; *Nat. Commun.*, 2023, 14, 368; *Angew. Chem., Int. Ed.*, 2022, 61, e202202200; *Angew. Chem., Int. Ed.*, 2022, 61, e202117347). Main-group sites have demonstrated limited reactivity concerning Fenton reactions, thereby contributing to the reduce in H_2O_2 decomposition.

Secondly, in our previous manuscript, we conducted quasi-*in situ* EPR measurements and H_2O_2 reduction reaction experiments to elucidate the Fenton and H_2O_2 RR activity of Pb SACs. As shown in Supplementary Fig. 35, no characteristic signals of hydroxyl radical resulting from Fenton reactions were discernible in the EPR spectra of Pb SA/OSC, Pb SA/OC, and OSC during the $2e^-$ ORR process. This finding led to the conclusion that the Pb SACs exhibited diminished activity towards Fenton reactions. Furthermore, the results of the H_2O_2 RR experiments indicated that the rate of H_2O_2 electroreduction on Pb SA/OSC exhibited only marginal increase with a higher overpotential and H_2O_2 concentration (Fig. 4f). The current density for H_2O_2 RR on Pb

SA/OSC remained below -0.07 mA cm^{-2} when the potential exceeded 0.40 V vs. RHE , indicating the limited activity of the main-group sites of Pb SA/OSC towards $\text{H}_2\text{O}_2\text{RR}$. Hence, it is evident that Pb SA/OSC played a pivotal role in avoiding the decomposition or electrochemical reduction of H_2O_2 .

Figure R4: (a) Quasi-*in situ* EPR spectra of Pb SA/OSC, Pb SA/OC, and OSC. (b) LSV of Pb SA/OSC in N_2 -saturated 0.10 M KOH electrolyte containing 1 mM , 5 mM , 10 mM , or $20 \text{ mM H}_2\text{O}_2$. (c) Schematics of the electrochemical cumulative H_2O_2 production. (The figures displayed here are intended to make our response easier to read, but the figures are already included in the previous manuscript)

Furthermore, minimal Fenton and $\text{H}_2\text{O}_2\text{RR}$ activity are beneficial for achieving a high net rate of H_2O_2 production (i.e., the production rate minus the electroreduction rate of H_2O_2), thereby facilitating H_2O_2 accumulation. As shown in Fig. 5c, the concentration of accumulated H_2O_2 increased nearly linearly, reaching 1358 mM ($4 \text{ wt.}\%$) after 2 h of electrolysis at 200 mA/cm^2 . Notably, the H_2O_2 FEs exhibited only a slight decrease trend (97% to 91%) throughout the course of electrolysis. These outcomes further underscore the efficacy of the Pb SA/OSC catalyst in preventing the depletion of generated H_2O_2 during electrolysis.

Finally, accepting the reviewer's suggestion, we have performed supplemented theoretical investigations to assess the stability of the synthesized H_2O_2 molecules. As demonstrated in the experimental results, the coordination structures of Pb sites exhibited various abilities in accumulating the synthesized H_2O_2 . To give a comprehensive comparative analysis of the impact of coordination environments, we took PbO_4 and PbS_4 as examples in the control group, while PbS_4O_2 as the target group. To probe the complex configurations of H_2O_2 within a solvation environment at the solid-liquid interface, we used the combined explicit-implicit water model to conduct a more in-depth study. Different initial geometries of H_2O_2 at the catalyst-water interface were constructed and relaxed. For the PbO_4 and PbS_4 catalysts, we observed the cleavage of the O-O bond in H_2O_2 concurrent with the formation of adsorbed *OH

of *O species on the Pb sites. In contrast, the PbS₄O₂ catalysts exhibited a capability to stabilize the molecular H₂O₂ through hydrogen bonding with water molecules. Therefore, the local coordination environment of the Pb SA/OSC catalyst, especially the PbS₄O₂ catalyst, played a crucial role in preserving H₂O₂ by inhibiting the O–O bond cleavage (For more details, please refer to our response in **Reply #3.1**).

To address the reviewer's concern, we have incorporated the related discussion in the revised manuscript (Page 21, Lines 392-397).

4. For stability testing, the authors did not give sufficient characterizations to prove the stability of catalyst after 100 h of electrolysis.

Reply #1.4: To verify the stability of the catalyst after 100 h of electrolysis, we examined the catalysts with transmission electron microscopy (TEM) X-ray diffraction (XRD) and Raman characterizations. Figure R5 shows the TEM images of the Pb SA/OSC at different magnifications after prolonged electrolysis, indicating the absence of Pb clusters or small PbS species. The XRD patterns (Figure R6) obtained from the Pb SA/OSC on GDE exhibited no distinctive peaks associated with crystalline PbS species after electrolysis, in alignment with the observation from the TEM measurements. Additionally, the Raman spectra (Figure R7) of the Pb SA/OSC revealed the characteristic D and G bands of conductive carbon materials, with calculated I_D/I_G values comparable before and after electrolysis. These findings demonstrate that there were no discernible structural changes in the Pb SA/OSC after the stability testing. Therefore, the Pb SA/OSC catalyst exhibited outstanding stability after 100 h of electrolysis.

To address the reviewer's concern, we have added the related description into the revised manuscript (Page 22, Lines 407-409), and incorporated the data presented in Figures R5-R7 into the revised Supplementary information (Figures S44-S46).

Figure R5: (a-d) TEM images at different magnifications of Pb SA/OSC obtained from

the GDE electrode after the stability test.

Figure R6: XRD patterns of Pb SA/OSC-GDE before and after the stability test.

Figure R7: Raman spectra of Pb SA/OSC-GDE before and after the stability test. The intensity ratios (I_D/I_G) of the D-band to G-band were calculated.

Response to Reviewer 2's comments

This study shows how Pb single sites coordinated with sulfur can increase the H₂O₂ selectivity. The study is unique in their approach with stitching together fundamental first principles studies, lab scale tests, a battery of characterisation studies, industrial scale tests, and an application of using the H₂O₂ for an oxidation reaction. The amalgamation of these factors create a unique study that is potentially of interest to Nature Communications. But, before that, a few technical points need to be revised, the manuscript in its current form is not suited for Nature Communications. Revising these points is critical to justify overall conclusions. The primary concerns are with explaining the rationale for selecting the DFT structures, not considering oxidized sulfur sites when evidence suggests otherwise, and not reporting the extent of sulfur leaching.

Reply: We sincerely appreciate the reviewer's insightful comments as well as the recognition and support of our work. In response to the reviewer's concerns, we have offered a comprehensive explanation regarding the selection of the DFT structures. We have also explored the impact of oxidized sulfur sites on the 2e⁻ ORR catalytic performances and reported the extent of sulfur leaching, all of which have significantly enhanced the depth and rigor of this work.

1. Figure 1 shows the structures used for DFT simulations. Why were oxidized sites not considered, when such oxidation of sulfur is expected and even seen experimentally (XPS studies, C-SO_x peaks)?

Reply #2.1: Following the reviewer's suggestion, we have examined the effect of oxidized sulfur (C-SO_x) groups on the PbS₄O₂ center. Since the linkage of the SO_x group to a C atom can vary (*Nat. Rev. Chem.*, 2023, 7, 573–589; *J. Am. Chem. Soc.*, 2023, 145, 30, 16835–16842), we selected a hexagonal ring structure of oxidized sulfur as a model to conduct DFT calculations (*J. Am. Chem. Soc.*, 2019, 141, 6254-6262). Given the larger spatial distance between C-SO_x group and Pb site, their interactions become weaker. To investigate the potential impact of C-SO_x group on catalytic activity, only one benzene ring was used to separate S and Pb. As shown in Figure R8a, the electron-withdrawing C-SO_x substitutes are located on the side C atom of the carbon support. Based on a set of intermediates (*OOH, *O and *OH, Figure R8b), the calculated overpotential for 2e⁻ORR pathway (pH = 0, U = 0 V) was 0.35 V, suggesting that the C-SO_x groups are unfavorable for the formation of H₂O₂ (Figure R8c). Here, the oxidized sulfur groups are not in the first coordination shell of Pb site, but in close proximity. They play a role similar to that of the enzyme pocket residues found in natural peroxidase during the catalytic process, including H₂O₂ binding and the cleavage of the O-O bond (*J. Am. Chem. Soc.*, 2023, 145, 30, 16835–16842).

As the primary goal of this work is to screen the first-shell coordination structure of the Pb site, a trade-off between computational cost and screening efficiency becomes essential. By focusing on the distinctions among various Pb sites, we assume that the effect of the C-SO_x group on the candidate Pb sites can be treated as a background

factor. However, this simplified approach may introduce uncertainty, including the spatial distance and the number of the C–SO_x group around the Pb site. Since the XPS detection of sulfur types is based on a statistical analysis of the entire sample, it is challenging to determine the specific spatial distribution of C–SO_x groups. Accepting the reviewer’s valuable suggestion, we have enhanced understanding of the effect of C–SO_x group on the oxygen reduction activity of Pb site by incorporating the analysis of the oxidized sulfur.

To address the reviewer’s concern, we have incorporated the data presented in Figure R8 into the revised Supplementary information (Figure S5).

Figure R8. Effect of the C–SO_x group on the oxygen reduction reaction. (a) DFT-optimized geometry of the C–SO_x group on the carbon support, denoted as PbS₄O₂-SO. (b) Intermediates geometries for the 2e⁻ORR pathway of PbS₄O₂-SO. (c) Volcano plots for the 2e⁻ and 4e⁻ ORR on various Pb SA/OSC.

2. What was the rationale for selecting these structures?

Reply #2.2: We select the DFT structures because of the following reasons:

Firstly, main-group SACs hold the potential to prevent the depletion of generated H₂O₂ during electrolysis by suppressing Fenton and H₂O₂ reduction reactions. For this potential, main-group sites are expected to surpass transition metals in terms of inert activity towards the electron transfer reactions, as they lack a combination of empty and occupied host orbitals. Given the demonstrated high activity of Pb materials in various electrocatalytic reactions, Pb-based catalysts were chosen as the main-group model catalysts. It is crucial to note that the performances in catalytic processes are highly dependent on the coordination environments due to electronic structure alterations. To tailor the 2e⁻ORR activity of Pb single-atom sites, a coordination environment engineering strategy was applied. Previous reports show that the strong chemical interaction between metal and sulfur atoms facilitates the formation of strong and thermally stable metal-sulfur bonding (*Nat. Commun.*, 2021, 12, 3135; *Sci. Adv.*, 2019, 5, eaax6322), significantly mitigating the aggregation of metal sites. Consequently, structure models of Pb SACs were constructed by supporting Pb atoms on graphene with simultaneous co-coordination of S and O atoms.

Subsequently, SACs with the typical M-N₄ structure have been extensively demonstrated to exhibit a high activity for various electrochemical reactions. Several potential Pb-X₄ models (where X = O, S) were constructed through diverse combinations of coordinated S and O atoms (PbO₄, PbS₁O₃, PbS₂O₂, PbS₃O₁, PbS₄). Moreover, PbS₃O₁-f, PbS₄-f models were also constructed to investigate the influence of S type within the graphene substrate. Accordingly, the adsorption energies of *OOH, *O and *OH on the Pb SACs structures were calculated. The results indicate that Pb SACs co-coordinated with both S and O and a higher S ratio, result in an optimal adsorption strength of the *OOH intermediate (Supplementary Fig. 2). Based on these predicted theoretical findings, we attempted to synthesize the Pb SACs coordinated with different S/O ratios.

Furthermore, it has been reported in the literature that the most common coordination numbers for main-group Pb sites are 4 and 6 (*J. Am. Chem. Soc.*, 2005, 127, 9495-9505; *Inorg. Chem.*, 1998, 37, 1853-1867). In this regard, a Pb SACs model was constructed in which Pb sites were co-coordinated by 6 S and/or O atoms with a higher S ratio (PbS₄O₂). Notably, PbS₄O₂ is positioned closest to the apex of the limiting potential volcano, akin to the PtHg₄ catalyst. As a result, PbS₄O₂ exhibits a high selectivity to for H₂O₂ formation with a low overpotential of 0.006 V. The EXAFS fitting of the optimized Pb SA/OSC catalyst clearly revealed that Pb sites are co-coordinated by both S and O atoms. The best-fit result of the EXAFS data (Supplementary Table 6) indicate that the coordination numbers of S and O for Pb sites closely resemble the PbS₄O₂ model. Additionally, the averaged interatomic distances derived from the fitting results also strongly concur with the distances of Pb–O and Pb–S in the PbS₄O₂ structure. This predicted structure is strongly supported by the experimental results.

To address the reviewer's concern, we have incorporated the related discussion into the revised manuscript (Page 6, Lines 102-104).

3. Can the free energy diagram be replotted at the equilibrium potential?

Reply #2.3: At pH = 0 and zero cell potential ($U = 0$), all elementary steps of oxygen reduction are exothermic for PbS₄O₂. However, when we shift the chemical potential of the electrons by an applied potential of 0.7 V (corresponding to the equilibrium potential for 2e⁻ oxygen reduction), PbS₄O₂ exhibits a flat free energy diagram for H₂O₂ production, suggesting a high catalytic activity with zero overpotential (Figure R9a). At the equilibrium potential of 1.23 V for 4e⁻ oxygen reduction, the PbS₄O₂ catalyst incorporates two uphill steps in the free energy diagram (Figure R9b), suggesting that the formation of H₂O₂ is more favored on PbS₄O₂ under these conditions. It should be noted that this thermodynamic analysis may not perfectly capture the complexity of the experimental results, which can be influenced by various kinetic and surface factors. Additionally, to provide a comprehensive perspective on the catalysts' performance, we included PbO₄ and PbS₄ as examples in the control group. The effect of various coordination environments on the catalytic performance at the equilibrium potential was examined (Figure R9c-f). At the corresponding equilibrium potentials of 2e⁻ and

$4e^-$ ORR pathways, no flat free energy diagram was observed for the control group. These comparisons underscore the unique attributes of PbS_4O_2 as a catalyst in $2e^-$ ORR electrochemical scenarios.

Figure R9. Free-energy diagrams of $2e^-$ and $4e^-$ pathways for oxygen reduction on (a, b) PbS_4O_2 , (c, d) PbS_4 and (e, f) PbO_4 catalysts in acidic medium at zero cell potential ($U = 0$), at the other potentials ($U = 0.2, 0.4, 0.7$ and 1.23 V).

4. Does the sulfur leach out? Pb leaching has been reported but what about other elements? Can pre/post characterisations be performed to confirm this point?

Reply #2.4: To find out whether sulfur leached out during the electrolysis, we conducted ICP-OES analysis after 6 h of continuous electrolysis. According to the element analysis of sulfur (Supplementary Fig. 12), the sulfur content in the Pb SA/OSC was determined to be approximately 2.2 wt%. The ICP-OES analysis revealed that the concentration of leached sulfur after H_2O_2 production reaction was approximately 0.011 mg L^{-1} , which accounted for 0.2 % of the total sulfur content only, thereby verifying the exceptional stability of the Pb SA/OSC catalyst.

To address the reviewer's concern, we have incorporated the related descriptions into the revised manuscript (Page 22, Lines 410-411).

5. Figure 4b: The three sites, Pb SA/OSC, Pb SA/OC and OSC all show faradaic

efficiencies in the same range. What are the error bars on the measurements? Are DFT calculations qualitatively accurate, especially using a simple limiting potential analysis, to differentiate the relative performance of these sites when their faradaic efficiencies vary by less than 10% at 0.6-0.7 V?

Reply #2.5: To accurately assess the faradaic efficiencies of the Pb SA/OSC, Pb SA/OC and OSC catalysts, we conducted multiple parallel tests (Figure R10). Specifically, we selected faradaic efficiencies at 0.3, 0.35, 0.4, 0.45, 0.5, 0.55, 0.6, 0.65, 0.7 V to calculate the error bars associated with the measurements. The small errors across multiple tests indicate the high reliability and repeatability of the faradaic efficiency measurements.

Figure R10: Multiple faradaic efficiency measurements of Pb SA/OSC, Pb SA/OC and OSC catalysts and faradaic efficiency measurements with error bars.

In addition, we agree with the reviewer about the limitations of DFT calculations in providing precise qualitative assessments of faradaic efficiency. DFT calculations are a commonly used tool for qualitatively simulating the properties of molecules and materials. In many cases, DFT calculations are useful for predicting trends, relative stabilities, and electronic structure properties. This enables experimentalists to try and make the material according to the information about the energetics of the chemical reactions (*Nat. Comput. Sci.*, 2022, 2, 539–541). However, it is important to note that the accuracy can vary depending on several factors, including the choice of exchange-correlation functional, basis set, and the system being studied. The limitation in

accuracy becomes more pronounced when dealing with systems with strong electron-electron correlation or long-range interactions. Considering the strengths and limitations of DFT, we have made informed decisions when using it in this work.

Specifically, we utilized DFT calculations to determine the relative thermodynamic selectivity of the candidate materials, which in turn informed the experimental adjustments to the S/O ratio for synthesizing Pb-based catalysts. For the coordination structure of Pb site (PbO_4 and PbS_4), which exclusively consists of either O or S atoms, DFT calculations indicated poor $2e^-$ -ORR performance, a finding confirmed by the corresponding control experiments. In contrast, when Pb site's coordination structure incorporates a mixture of O and S atoms (PbS_1O_3 , PbS_2O_2 , PbS_3O_1 , and PbS_4O_2), the predicted performances depend on the various S/O ratios. Notably, a S/O ratio of 2:1 exhibits optimal performance (PbS_4O_2), as validated by experiments. Therefore, instead of relying solely on absolute overpotential values and expecting exact alignment with experimentally measured values, we utilized the catalytic trends from DFT calculations and combined with the subsequent material synthesis to achieve high $2e^-$ -ORR performance.

6. In the SI, the authors state that the adsorption configurations of $\text{O}^*/\text{OH}^*/\text{OOH}^*$ were determined after searching across all possible configurations. Please report the configurations scanned and the adsorption energies determined in the SI, in addition to reporting the most stable structures.

Reply #2.6: For the Pb site of PbS_4O_2 , we incremented the adsorption angles of ORR intermediates in the horizontal plane from 0° to 360° with a step size of 60° (Figure R11b). Such an exploration of adsorption orientations allowed us to assess the potential adsorption behavior. Started from a range of initial structures, the adsorption geometries for $^*\text{OOH}$ were relaxed to five orientations (Figure R11c), while the geometries for $^*\text{OH}$ were optimized to two orientations (Figure R11d). The free energies for the adsorption of $2e^-$ -ORR intermediates were calculated and are plotted in Figure R11a, resulting an overpotential ranging from 0.11 to 0.17 V, which is larger than the reported value. We acknowledge the discrepancy in the observed overpotential range. However, while screening the coordination structures of Pb sites, we have concurrently observed that the fluctuation in $2e^-$ -ORR overpotential on the PbS_4O_2 catalyst remained notably smaller than that of other catalysts (0.95 V for PbS_4). This result substantiates that the $2e^-$ -ORR performance of the PbS_4O_2 catalyst surpassed those of other screened catalysts.

Figure R11. Calculated free energies of adsorption of ORR intermediates on PbS_4O_2 . (b) Rotation states of the initial adsorbates ($*OH$ and $*OOH$). (c, d) The corresponding DFT-optimized structures of $*OH$ and $*OOH$, respectively.

7. The captions in the SI need more detail so that they can stand-alone with the figure. Please expand on the captions in the SI describing key aspects of the figure. This change will improve readability.

Reply #2.7: Accepting the reviewer's suggestion, we have expanded on the captions describing key aspects of the figures in the revised Supplementary information, which significantly improves overall readability.

8. What are the vibrational frequencies used to determine the ZPE? Were solvation effects included? Please report these aspects for greater reproducibility.

Reply #2.8: The vibrational frequencies used to determine the Zero-Point Energy (ZPE) were supplemented in Table R1. ZPE is calculated as: $ZPE = 0.5hc\sum\nu NA$, where h is the Planck's Constant, c is the speed of light in m/s, NA is the Avogadro's Number, $\sum\nu$ is the sum of vibrational frequencies of adsorbates.

The reviewer is correct as the influence of solvent environment on the electrochemical catalysis is important, especially when considering the adsorption configurations at the solid–liquid interface in comparison to a vacuum model. There are two main methods to add solvation within DFT: the explicit approach and the implicit approach (*ACS Catal.*, 2019, 9, 2, 920–931). The explicit approach involves adding water molecules into the system, offering superior descriptions of solvation and field effects, but it comes with significant computational costs for exploring configuration space (*J. Chem. Theory Comput.*, 2019, 15, 12, 6895–6906). In contrast, the implicit

approach utilizes the response of a continuum dielectric to model the solvation effect, offering the flexibility to conduct constant potential calculations by adjusting the charge at the interface, but these benefits typically come at the cost of a precise representation of solvation effects and interfacial capacitance (*Chem. Rev.*, 2022, 122, 12, 10777–10820). However, studying surface chemical reactions with solvation effects remains challenging due to the complexity of models and high computational costs (*Proc. Natl. Acad. Sci. U. S. A.* 2017, 114, 1795–1800). Nevertheless, we have combined the explicit and implicit water models to investigate the effect of solvent molecules on the stability of the synthesized H₂O₂ at the electrochemical interface (Figures R12-14). In order to probe the solvent environment across various structures of the solids–liquid interface, we took PbO₄ and PbS₄ as examples in the control group, with PbS₄O₂ in the target group. More details can be found in our response in **Reply #3.1**.

To address the reviewer’s concern, we have incorporated Table R1 into the revised Supplementary information as Supplementary Table 5 and the data presented in Figures R12-R14 into the revised Supplementary information (Figures S36-S38).

Table R1. Vibrational frequencies used to determine the zero-point energy correction (ZPE) for adsorbates (*OOH, *O and *OH) at the Pb sites of various catalysts at T = 298K.

Catalyst	PbS ₃ O _{1-f}	PbS _{4-f}	PbO ₄	PbS ₁ O ₃	PbS ₂ O ₂	PbS ₃ O ₁	PbS ₄	PbS ₄ O ₂
	3513.03	3631.8	3447.59	3634.48	3639.69	3637.03	3431.78	3557.06
	1344.22	1303.86	1302.07	1329.53	1267.21	1334.13	1329.1	1315.82
ν_{*OOH} (cm ⁻¹)	802.28	802.65	1252.33	809.83	830.03	862.5	800.33	878.38
	605.9	469.97	549.21	429.45	454.92	454.02	586.01	433.61
	399.85	368.07	283.12	316.14	250.86	359.68	373.91	254.9
	112.17	238.39	108.35	82.22	72.16	71.06	108.33	136.14
$\Sigma \nu_{*OOH}$ (cm ⁻¹)	6777.45	6932.61	6942.68	6601.66	6514.88	6718.42	6629.46	6575.91
ZPE _{*OOH} (eV)	0.42	0.43	0.43	0.41	0.4	0.42	0.41	0.41
ν_{*O} (cm ⁻¹)	684.83	697.2	1255.52	661.8	770.21	597.57	693.21	614.14
	164.24				181.61		35.58	
$\Sigma \nu_{*O}$ (cm ⁻¹)	849.08	697.2	1255.52	661.8	951.82	597.57	728.79	614.14
ZPE _{*O}	0.05	0.04	0.08	0.04	0.06	0.04	0.05	0.04

(eV)								
	3742.23	3825.75	3829.68	3762.94	3807.04	3474.01	3718.68	
$\nu_{*OH}(\text{cm}^{-1})$	646.65	660.99	626.27	839.69	555.35	703.55	686.01	3648.81
1)	416.27	357.69	403.26	354.47	431.9	515.08	411.1	569.01
	371.09	300.62	255.75	222.42	197.26	378.87	378.35	362.27
$\Sigma\nu_{*OH}$								
(cm^{-1})	5176.24	5145.05	5114.96	5179.52	4991.54	5071.51	5194.15	4580.1
ZPE $_{*O}$								
(eV)	0.32	0.32	0.32	0.32	0.31	0.31	0.32	0.28

Figure R12. DFT-optimized geometries of H_2O_2 molecule at the solid–liquid interface. (a–c) Solvated H_2O_2 and the production of $*\text{OH}$ intermediates on the PbO_4 catalyst, (d–f) H_2O_2 and the $*\text{O}$ intermediates at the PbS_4 catalyst.

Figure R13. Charge transfer behavior and molecular bond strength relationship. The charge density differences for H₂O₂ adsorbed on (a) PbO₄ and (c) PbS₄. The yellow color in the representation signifies a high electron density, while the blue color indicates scarce electron density. Electron density difference maps of 0.008 e⁻/bohr³. The crystal orbital Hamilton populations (COHP) of (b) *OH adsorbed on PbO₄ and (d)*O adsorbed on PbS₄ after H₂O₂ activation.

Figure R14. DFT-optimized geometries of H₂O₂ molecule at the solid–liquid interface. (a-c) Solvated H₂O₂ with two different adsorption structures on the PbS₄O₂ catalyst, (d, e) Calculated charge density differences for H₂O₂ on PbS₄ O₂ (isosurface value=0.008 e⁻/bohr³).

9. Line 188, Main text etc: Please round all adsorption energies calculated using DFT to two decimal places keeping in mind the general numerical accuracies of these simulations.

Reply #2.9: Accepting the reviewer’s suggestion, we have updated the related descriptions in the revised manuscript and the Supplementary information (Supplementary Tables 3 and 4).

Response to Reviewer 3's comments

This work demonstrates the synthesis of a Pb SA/OSC catalyst, which exhibited relatively promising H₂O₂ electrosynthesis activity and selectivity with remarkable stability (100h) in an alkaline electrolyte. The Pb SA/OSC catalyst was applied to a more practical gas diffusion electrode-based reactor and a high current density of 400 mA cm⁻² was achieved.

This catalyst is interesting and has not been reported for 2-e ORR so far. However, a few technical issues must be fully addressed to meet the high standard of Nature Communications. Detailed comments are listed below:

Reply: We highly value the reviewer's insightful comments and valuable suggestions, which have notably improved the quality of our manuscript. In this revised version of the manuscript, we have diligently addressed all the questions and concerns raised by the reviewer.

1. Some recent advancements in theoretical understanding and material innovation have led to the development of a series of efficient main group metal single-atom catalysts for the electrosynthesis of H₂O₂ in alkaline media (Angew. Chem. Int. Ed.2022, 134, e202117347). In comparison with the activity, the Pb SA/OSC catalyst showed excellent stability in the flow cell. More theoretical discussion is needed to explain this advantage.

Reply #3.1: The remarkable work (Angew. Chem. Int. Ed., 2022, 134, e202117347) set an important research paradigm for the development of main-group SACs through a combination of theoretical understanding and material innovation. This excellent work has been thoroughly read and cited in our previous manuscript.

In the work, we utilized theoretical simulations to investigate the impact of coordination environment of main-group sites on the electrocatalytic activity. Since SACs with the well-defined M-N₄ structure have been extensively demonstrated to exhibit a high activity for various electrochemical reactions, we constructed Pb-X₄ models (where X = O, S) using various combinations of coordinated S and O atoms (PbO₄, PbS₁O₃, PbS₂O₂, PbS₃O₁, PbS₄, PbS₃O_{1-f}, PbS_{4-f}). Additionally, it has been reported in the literature that the most common coordination numbers for main-group Pb sites are 4 and 6 (J. Am. Chem. Soc., 2005, 127, 9495-9505; Inorg. Chem., 1998, 37, 1853-1867). Therefore, we also constructed a Pb SACs model with 6 coordination atoms (PbS₄O₂). Based on the theoretical results, we found that Pb SA/OSC (PbS₄O₂), co-coordinated with both S and O and with a higher S ratio, is positioned closest to the apex of the limiting potential volcano, exhibiting a high selectivity for H₂O₂ formation. Inspired by these theoretical results, we endeavored to synthesize the Pb SACs coordinated with different S/O ratios experimentally. Notably, the EXAFS fitting of the optimized Pb SA/OSC catalyst revealed that the coordination numbers of S and O for Pb sites, as well as the distances of Pb-O and Pb-S, closely resemble the PbS₄O₂ model. Therefore, guided by the DFT calculations, we successfully developed main-group Pb SACs with a high intrinsic H₂O₂-producing selectivity.

The stable performance of the flow-cell system was demonstrated via the continuous and stable production of H₂O₂ over 100 h (Fig. 5f). Under realistic conditions, the effective long-term accumulation of H₂O₂ resulted from a balance between the generation and decomposition of H₂O₂ molecules at the electrode–liquid interface. Accepting the reviewer's suggestion, we have supplemented theoretical investigations. Here, molecular H₂O₂ not only interacts with the main-group site of Pb SA/OSC catalyst, but also communicates with the solvent molecules. To probe the complex configurations of H₂O₂ within a solvation environment, we used the combined explicit-implicit water model to conduct a more in-depth study. Different initial geometries of H₂O₂ at the solid–liquid interface were constructed and relaxed. To compare the various coordination environments of the catalysts, we took PbO₄ and PbS₄ as examples in the control group, while PbS₄O₂ as the target group.

At the surface of PbO₄ catalyst, the molecular state of H₂O₂ was co-adsorbed with two H₂O on the Pb site, forming hydrogen bonds with the surrounding water molecules (Figure R12a), resulting in a decrease in the corresponding Pb–O distance from the initial 3.28 Å to 2.56 Å. While the decomposition of H₂O₂ was more thermally favored (-2.07 eV) with the breaking of the O–O bond, resulting in one adsorbed *OH species and one free •OH species (*H₂O₂ → *OH + •OH, Figure R12b). The *OH species can be further reduced to generate H₂O (*Phys. Chem. Chem. Phys.*, 2013, 15, 148-153). Similarly, at the surface of PbS₄ catalyst, H₂O₂ was also co-adsorbed with two H₂O molecules (Figure R12d), resulting in a decrease in the Pb–O distance from the initial 3.07 Å to 2.57 Å. While the decomposition of H₂O₂ was more favored (-2.13 eV) with the cleavage of the O–O bond, generating one adsorbed *O species and one H₂O molecule (*H₂O₂ → *O + H₂O, Figure R12e). Therefore, the activation process of H₂O₂ on the control samples was favored (Figure R12c and f). As a result, *OH-*H₂O species were adsorbed at the PbS₄ catalyst and the *O-*2H₂O adsorbed at PbO₄, respectively. As shown in Figure R13, the charge density enriched at the region between the Pb atom and the dissociated O atom from H₂O₂, forming a chemical bonding of Pb–O. Integral crystal orbital Hamilton populations (ICOHP) results show that the intensity of the new Pb–O bond when H₂O₂ was adsorbed on PbO₄ (-ICOHP = 0.97) was close to that on PbS₄ (-ICOHP = 1.01). In contrast, at the surface of PbS₄O₂ catalyst, the molecular H₂O₂ can be stabilized through hydrogen bonding with the water molecules (Figure R14). The O–O distance (1.47 Å) in adsorbed H₂O₂ was close to the value in molecular H₂O₂. While the Pb–O distance increased to 4.08 Å from the initial value of 3.28 Å. The energy difference of 0.7 eV between the two geometries of adsorbed H₂O₂ can be attributed to the variation in their initial solvation structures. Hence, the coordination environments of the Pb SA-based catalysts played a critical role in inhibiting the decomposition of H₂O₂.

Additionally, the Pb SA/OSC catalyst with the unique super-coordinated structure (PbS₄O₂) exhibited robust tolerance to S-containing contaminant poisoning, which is advantageous for long-term stability during practical applications (More details can be found in our response in **Reply #3.4**). Specifically, the Pb SA/OSC catalyst demonstrated a notable capability to stabilize and accumulate H₂O₂ at the solid–liquid

interface, as confirmed by the long-term flow-cell experiments. The data presented in Figures R12-R14 have been incorporated into the revised Supplementary information (Figures S36-S38).

Figure R12. DFT-optimized geometries of H₂O₂ molecule at the solid–liquid interface. (a-c) Solvated H₂O₂ and the production of *OH intermediates on the PbO₄ catalyst, (d-f) H₂O₂ and the *O intermediates at the PbS₄ catalyst. (The Figure secondly shown here is intended to make the Reply easier to read, but the Figure is still numbered as Figure R12 as above)

Figure R13. Charge transfer behavior and molecular bond strength relationship. The charge density differences for H₂O₂ adsorbed on (a) PbO₄ and (c) PbS₄. The yellow color in the representation signifies a high electron density, while the blue color indicates scarce electron density. Electron density difference maps of 0.008 e⁻/bohr³. The crystal orbital Hamilton populations (COHP) of (b) *OH adsorbed on PbO₄ and (d) *O adsorbed on PbS₄ after H₂O₂ activation. (The Figure secondly shown here is intended to make the Reply easier to read, but the Figure is still numbered as Figure R13 as above)

Figure R14. DFT-optimized geometries of H₂O₂ molecule at the solid–liquid interface. (a-c) Solvated H₂O₂ with two different adsorption structures on the PbS₄O₂ catalyst, (d, e) Calculated charge density differences for H₂O₂ on PbS₄O₂ (isosurface value=0.008 e⁻/bohr³). (The Figure secondly shown here is intended to make the Reply easier to read, but the Figure is still numbered as Figure R14 as above)

2. In Figure 1c, the DFT calculations on the Gibbs free energy should be reconducted to consider the related species in alkaline media (OH⁻ and HO₂⁻) instead of those in acid (H₂O and H₂O₂). The authors could refer to some references, such as *Phys. Chem. Chem. Phys.*, 2013, 15, 148-153.

Reply #3.2: Accepting the reviewer’s suggestion, we have calculated the free-energy diagrams for catalysts in alkaline media. In the recommended reference (*Phys. Chem. Chem. Phys.*, 2013, 15, 148-153) the mechanism of oxygen reduction reaction (ORR) on Co–N_x (x = 2, 4) electrocatalysts in both alkaline and acidic media was investigated. It has proven to be highly beneficial for our work. The complete four-electron reaction in alkaline medium is composed of the following elementary steps:

The complete two-electron reaction in alkaline medium is composed of the following elementary steps:

Therefore, the calculated computed free energy diagrams (Figure R15) for the catalysts in alkaline medium were supplemented. At the surface of PbS_4O_2 catalyst, formation of $*\text{OOH}$ remains uphill for $U > 0$ V. All elementary reaction steps for $*\text{OOH}$ to H_2O_2 (2e^- -pathway) are downhill by applying low potential ($U = 0.2$ V), while the generation of H_2O (4e^- -pathway) is downhill at the corresponding equilibrium potential ($U = 0.4$ V). In contrast, at the surface of PbS_4 or PbO_4 , reduction of $*\text{OOH}$ to H_2O_2 or H_2O is uphill processes at high potential.

To address the reviewer's concern, we have updated Figure 1c in the revised manuscript, supplemented Figure R15 into the revised Supplementary information (Supplementary Fig. 4). Besides, we have cited the excellent work (*Phys. Chem. Chem. Phys.*, 2013, 15, 148-153) in the revised manuscript.

Figure R15. Free-energy diagrams of 2e^- and 4e^- pathways for oxygen reduction on (a, b) PbS_4O_2 , (c, d) PbS_4 and (e, f) PbO_4 catalysts in alkaline medium at zero cell potential ($U = 0$), at the other potentials ($U = 0.2, 0.4, 0.7$ and 1.23 V).

3. Operando FTIR Spectroscopy of Pb SA/OC and OSC should be offered to prove the adsorption of O_2 and OOH on Pb SAs rather than S and O. Also, isotope experiment is

required to confirm the origin of OOH.

Reply #3.3: To investigate the adsorption sites of the prepared catalysts for O₂ and OOH, we have applied operando FTIR spectroscopy to examine Pb SA/OC and OSC. As shown in Figure R16, potential-dependent absorption bands were observed in the FTIR spectra of both Pb SA/OC and OSC, indicating that the *OOH mediated 2e⁻ ORR pathway on these catalysts. Notably, the absorption bands of Pb SA/OC (1238 cm⁻¹) and OSC (1243 cm⁻¹), associated with *OOH, exhibited distinct shifts compared to that of Pb SA/OSC (1254 cm⁻¹). This observation implies that the adsorption sites for *OOH in Pb SA/OSC were Pb single atoms. On the other hand, the adsorption sites of Pb SA/OC and OSC were different and likely to involve non-metallic S, O or C sites. Indeed, an exemplary study (*J. Am. Chem. Soc.*, 2021, 143, 7819–7827) demonstrated that the C atom adjacent to the coordinated O atom served as the adsorption site (in the case of CoO₄ SACs) for *OOH. These analyses indicate that the adsorption sites of Pb SA/OSC for O₂ and OOH were the Pb single atoms (Pb SAs).

Figure R16: (a, b) *In-situ* ATR-SEIRAS spectra collected on the OSC and Pb SA/OC catalysts in O₂-saturated 0.10 M KOH catholyte.

To further analyze the origin of the *OOH band, we have conducted an *in-situ* FTIR experiment using D₂O instead of H₂O as the solvent. The results of the isotopic-labeling study reveal that the vibration band of *OOH underwent a downshift to 1234 cm⁻¹ in the deuterated medium (Figure R17). Similar shift results have been reported in previous studies regarding operando FTIR spectroscopy of the *OOH intermediate (*ACS Catal.*, 2022, 12, 5345–5355; *J. Phys. Chem. B*, 2005, 109, 16563–16566). These findings strongly suggest the involvement of hydrogen atoms in the vibration mode at 1254 cm⁻¹, thus confirming the origin of *OOH.

To address the reviewer's concern, we have added the above discussion into the revised manuscript (Page 18, Lines 328-332), and incorporated Figures R16 and R17 into the revised Supplementary information as Figures S31 and S32.

Figure R17: *In-situ* ATR-SEIRAS spectra collected on the Pb SA/OSC catalyst in an O₂-saturated 0.10 M KOH solution and a deuterated medium.

4. Thiocyanide (SCN⁻) poisoning experiment is needed to probe the active site of Pb SA/OSC.

Reply #3.4: Thiocyanate ions (SCN⁻) exhibit a strong binding affinity to metal sites, facilitating the selective blockage of intermediate adsorption on these metal sites. Therefore, SCN⁻ poisoning experiments are commonly used to identify the active sites of single-atom catalysts.

Firstly, we conducted a poisoning experiment by collecting the RDE curve of Pb SA/OSC in the presence of thiocyanate ions. Remarkably, the 2e⁻ ORR activity of Pb SA/OSC remained almost unchanged upon the addition of 0.1 M KSCN (Figure R18), indicating that Pb SA/OSC exhibited a robust tolerance to SCN⁻ poisoning. The slight increase in current observed in the presence of KSCN was attributed to the elevated electrolyte concentration. We hypothesize that the resilience to SCN⁻ was likely associated with the unique super-coordinated structure (PbS₄O₂) of Pb SA/OSC, where the first coordination was modulated by S and O. Given that the main-group Pb sites were already coordinated by four S atoms, it is not surprising that S-containing SCN⁻ showed negligible effects on the 2e⁻ ORR activity of Pb SA/OSC.

Secondly, the DFT calculations reveal that the formation of S and O super-coordinated Pb moieties was the origin of the exceptional 2e⁻ ORR selectivity of Pb SA/OSC. Moreover, based on the RRDE measurements, the Pb SA/OSC exhibited a superior 2e⁻ ORR activity compared to OSC without Pb sites. Therefore, the main-group Pb atom served as the active site of Pb SA/OSC.

To address the reviewer's concern, we have incorporated the information presented in Figure R18 into the revised Supplementary information (Figure S29).

Figure R18: RDE curves of Pb SA/OSC in O₂-saturated 0.10 M KOH electrolyte with (red lines) and without (black lines) 0.1 M KSCN.

5. To demonstrate the high stability of the Pb SA/OSC catalyst, some post characterizations are required.

Reply #3.5: To verify the stability of the catalyst after 100 h of electrolysis, we examined the Pb SA/OSC catalyst with the transmission electron microscopy (TEM) X-ray diffraction (XRD) and Raman. Figure R8 shows the TEM images of the Pb SA/OSC at different magnifications after prolonged electrolysis, indicating the absence of Pb clusters or small PbS species. The XRD patterns (Figure R9) obtained from Pb SA/OSC on GDE exhibited no distinctive peaks associated with crystalline PbS species after electrolysis, in alignment with the observation from the TEM measurements. Additionally, the Raman spectra (Figure R10) of the Pb SA/OSC revealed the characteristic D and G bands of conductive carbon materials, with calculated I_D/I_G values comparable before and after electrolysis. These findings demonstrate that there were no discernible structural changes in Pb SA/OSC after the stability testing. Therefore, the Pb SA/OSC catalyst exhibited an outstanding stability after 100 h of electrolysis.

To address the reviewer's concern, we have added the related descriptions in the revised manuscript (Page 22, Lines 407-409), and incorporated the data in Figures R5-R7 into the revised Supplementary information (Figures S44-S46).

Figure R5: (a-d) TEM images at different magnifications of the Pb SA/OSC obtained from the GDE electrode after the stability test. (The figure shown here is intended to make the response easier to read, but the figure is still numbered as Figure R5 as above)

Figure R6: XRD patterns of Pb SA/OSC-GDE before and after the stability test. (The figure shown here is intended to make the response easier to read, but the figure is still numbered as Figure R6 as above)

Figure R7: Raman spectra of Pb SA/OSC-GDE before and after the stability test. The intensity ratios (I_D/I_G) of the D-band to G-band were calculated. (The figure shown here is intended to make the response easier to read, but the figure is still numbered as Figure R7 as above)

6. The structural information was gained from the EXAFS fitting analysis, and the fitting errors must be provided.

Reply #3.6: We have provided the EXAFS fitting errors in Supplementary Table 6. “Error bounds that characterize the structural parameters obtained by EXAFS spectroscopy were estimated as $N \pm 20\%$; $R \pm 1\%$; $\sigma^2 \pm 20\%$; $\Delta E_0 \pm 20\%$ ”.

REVIEWERS' COMMENTS

Reviewer #1 (Remarks to the Author):

I am satisfied with the revision. This paper is suitable for publication in the present version.

Reviewer #2 (Remarks to the Author):

The authors have addressed all questions by the reviewers and the manuscript can be accepted.

Reviewer #3 (Remarks to the Author):

The authors have cleared all of the questions raised in this revision. Our concerns, such as the theoretical discussion on the stability of Pb SA/OSC catalyst and DFT calculations, have been successfully addressed. The quality of this manuscript has been improved, and we have no further comments.

REVIEWERS' COMMENTS:

Reviewer #1 (Remarks to the Author):

I am satisfied with the revision. This paper is suitable for publication in the present version.

Reply: We really thank the reviewer for his/her positive comments.

Reviewer #2 (Remarks to the Author):

The authors have addressed all questions by the reviewers and the manuscript can be accepted.

Reply: Thank you for the nice comments.

Reviewer #3 (Remarks to the Author):

The authors have cleared all of the questions raised in this revision. Our concerns, such as the theoretical discussion on the stability of Pb SA/OSC catalyst and DFT calculations, have been successfully addressed. The quality of this manuscript has been improved, and we have no further comments.

Reply: We really thank the reviewer for his/her positive comments.